# Same-Expert Iteration Improves a Translation MoE Where Expert Communication Does Not

## Abstract

Mixture-of-Experts (MoE) models achieve scalability through sparse expert routing, but experts process tokens independently. A natural hypothesis is that enabling expert communication—through learned topologies, message passing, or sequential chains—should improve performance. We test this hypothesis on WMT14 En-De translation with a small decoder-only Transformer, evaluating ten communication approaches across seven experimental axes. We find no clear evidence that any variant improves over standard MoE, though modest sample sizes limit power for detecting small effects; several variants degrade validation perplexity. We hypothesize that explicit message passing between parallel experts may duplicate the implicit mixing that the router's weighted output combination already performs on the same per-token inputs. We then study a mechanism that instead changes the input to the expert computation: applying the *same* router-selected expert twice in sequence. This modification—**expert depth**—achieves 7.4% lower validation perplexity at matched FLOPs and equal epochs, consistently across five seeds; a wall-clock-matched baseline trained longer reaches lower perplexity, so this is a FLOP-efficiency result rather than a wall-clock one. Capacity ablations attribute the gain to the second pass rather than added width; mechanistically, the second pass acts as a contracting, anti-aligned correction rather than fixed-point refinement; and dense controls indicate that the benefit of depth at this scale is not MoE-specific. At a doubled FLOP budget, however, depth stacks productively with top-2 routing, outperforming a deeper dense model matched in FFN FLOPs, while adding inter-expert messages to this strongest configuration yields no benefit in a matched-seed control—consistent with the mixing-redundancy hypothesis. In this translation setting, the value of sparse routing appears to lie in combining expert selection and diversity with deeper per-expert processing, rather than in explicit inter-expert communication.

## 1 Introduction

Sparse Mixture-of-Experts (MoE) models have emerged as a powerful approach for scaling model capacity without proportional increases in compute (Jacobs et al., 1991; Shazeer et al., 2017; Fedus et al., 2022; Lepikhin et al., 2021; Jiang et al., 2024). In a typical MoE layer, the dense feed-forward block of a Transformer is replaced by $N$ "expert" feed-forward networks together with a learned router: for each token, the router scores all $N$ experts, only the top-$k$ (typically 1 or 2) are evaluated, and their outputs are combined via a weighted sum. Because only $k$ of $N$ experts run per token, total parameter count can grow with $N$ while per-token compute stays roughly constant—the property that makes MoE attractive at scale. Significant efforts have been directed toward improving routing mechanisms, load balancing, and expert specialization, but the experts themselves remain independent: each processes its assigned tokens in isolation, with no mechanism for inter-expert coordination.

A natural question is whether this independence is a limitation. Several recent works have explored mechanisms for **expert communication**—enabling experts to exchange information through message passing, learned topologies, or sequential composition (Wang et al., 2025). The intuition is compelling: if experts could coordinate, they might avoid redundant computation, share specialized knowledge, or produce more coherent outputs. However, this intuition has not been systematically tested: prior work typically proposes

a single communication mechanism and evaluates it in isolation, without ablating the design axes that could affect the outcome.

In this work, we evaluate expert communication in MoE on a translation benchmark. We test ten approaches across seven experimental axes—including communication timing (before vs. after expert processing), topology structure (learnable, random, and fully-connected topologies, plus topology-governed chains), expert count (8, 16, 32), training duration, domain diversity, and sequential composition—on WMT14 English-German translation with a ~290M-parameter Transformer (32 experts per layer; ~40M parameters active per token). All experiments, and therefore all claims in this paper, concern this single setting—a small decoder-only model on one translation benchmark; we make this scope explicit throughout and return to it in §9.

We find no clear evidence that any communication variant improves over the baseline: no configuration tested with 3+ seeds shows a significant improvement (all nominal improvements have $p > 0.6$), and six screening experiments (1–2 seeds) are consistent with this pattern. We note that these sample sizes provide limited power to detect small effects ($< 1\%$), so we frame these as absence of clear evidence rather than evidence of absence. Several communication variants increase validation perplexity. Notably, models *learn to use* the communication mechanisms—gates open from their suppressive initialization, learned topologies grow in density, and gradients flow through message-passing parameters—yet validation perplexity does not improve. We hypothesize that a structural property of standard MoE contributes: within a single token's forward computation, all selected experts receive the same input, so explicit inter-expert messages may be largely redundant with the **implicit mixing** that the router's learned weighted output combination already performs (§4). Even sequential chains with different experts (Wang et al., 2025), which create genuine asymmetry by letting each expert see its predecessor's output, show no significant gain in our setting.

These negative results led us to an unexpected discovery. While investigating sequential expert chains, we found that applying the *same* expert twice—iterative application rather than diverse composition—produces the only statistically significant improvement in the entire study. We call this approach **expert depth**: the router selects a single expert, which processes the token and then processes its own output. Figure 1 contrasts the three paradigms. Expert depth achieves 7.4% lower validation perplexity than standard MoE at matched FLOPs and equal training epochs, an effect that is consistent across all five random seeds and grows monotonically over training, accompanied by markedly better expert load balance. The two passes run sequentially, so the gain carries a wall-clock cost that we quantify explicitly (§6.1).

To understand whether this improvement reflects genuine iterative refinement or simply increased per-token computation, we conduct a four-way ablation (§6.2). A standard single expert and a width-doubled expert with twice the parameters produce identical perplexity, suggesting that added capacity alone does not explain the improvement. The observed gain appears attributable to the second pass through the same expert. Additional dense and depth-×-diversity controls (§6.3) and a mechanism analysis (§7) sharpen this picture, as summarized in the contributions below.

Overall, our contributions are four-fold:

- We evaluate ten approaches to expert communication in MoE across seven experimental axes; no configuration tested at 3+ seeds shows a statistically significant improvement in this setting (§3). We propose a structural hypothesis: explicit message passing may be redundant with the implicit mixing the router already performs (§4).

- We introduce **expert depth**, a simple modification that applies the same router-selected expert twice, achieving 7.4% lower perplexity at matched FLOPs and equal epochs ($p=0.0025$, 5 seeds), and we state its compute scope precisely: a FLOP-efficiency result, not a wall-clock one (§5, §6.1).

- Capacity ablations, dense controls, and a mechanism analysis isolate what drives the improvement: width provides no benefit, the second pass—a contracting, anti-aligned correction—accounts for the gain, and a FLOP-matched deeper dense model gains comparably, so depth at this scale is not MoE-specific (§6.2, §6.3, §7).

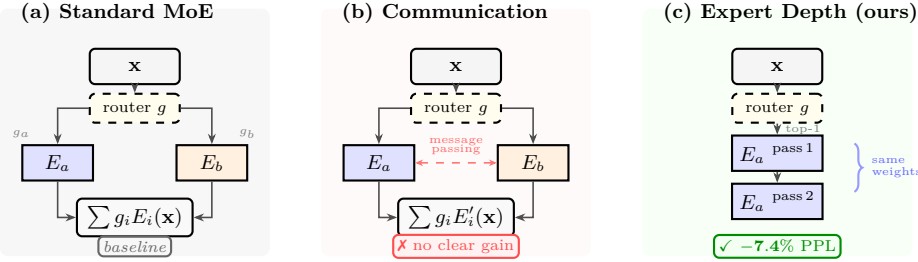

Figure 1: Three paradigms for expert computation in MoE. **(a)** Standard parallel routing. **(b)** Expert communication via message passing—10 variants tested across 1–3 seeds each; no configuration tested with 3+ seeds shows a statistically significant improvement, and six screening experiments are directionally consistent (§3). **(c)** Expert depth: the same expert processes its own output, the only approach showing a statistically significant improvement in this study (5 seeds, $p=0.0025$; §5).

- At a doubled FLOP budget, we show that depth and routing diversity stack: applying both top-2 experts twice outperforms a FLOP-matched 16-layer dense model, while inserting inter-expert messages into this strongest configuration changes nothing—extending the negative communication result to a compute-rich regime (§6.3).

## 2 Background and Experimental Setup

In a standard MoE layer (Shazeer et al., 2017; Vaswani et al., 2017), each token $\mathbf{x} \in \mathbb{R}^d$ is processed by a subset of $N$ expert modules $\{E_1, \ldots, E_N\}$ selected by a learned router $g$:

$$\mathbf{y} = \sum_{i \in \mathcal{S}(\mathbf{x})} g_i(\mathbf{x}) \cdot E_i(\mathbf{x}), \quad \mathcal{S}(\mathbf{x}) = \text{top-}k\big(g(\mathbf{x}), k\big) \tag{1}$$

where each expert $E_i$ is a two-layer FFN (linear projection → activation → linear projection), $g(\mathbf{x})$ produces routing scores via a learned projection followed by softmax, and $\mathcal{S}(\mathbf{x})$ selects the top-$k$ experts by score. The output is a weighted combination of the selected experts' outputs, providing implicit coordination through soft-ensembling. Throughout the paper we distinguish two forms of sequential expert computation: composition with *different* experts, $E_b(E_a(\mathbf{x}))$, as proposed by Chain-of-Experts (Wang et al., 2025), and iteration of the *same* expert, $E_a(E_a(\mathbf{x}))$, which we call *expert depth*. Our negative findings cover different-expert composition and parallel message passing; our positive finding concerns same-expert iteration.

All experiments use an identical Transformer decoder (Vaswani et al., 2017) with $\sim$290M total parameters ($\sim$268M expert weights; $\sim$40M active per token), 4 layers, hidden dimension 512, and 8 attention heads. Each layer contains 32 FFN experts with hidden dimension 2048 and top-$k$=2 routing. We train on WMT14 English-German translation using commoncrawl and europarl as training data (4.3M sentence pairs) and the news-commentary-v9 corpus as a held-out validation set (201K pairs). This is a custom split derived from WMT14 data: news-commentary is a distinct corpus from the training sources, though it is not the standard WMT14 benchmark test set (newstest2014, which we use separately for BLEU evaluation). We use shared BPE tokenization (Sennrich et al., 2016) and train for 5 epochs using AdamW (Kingma & Ba, 2015; Loshchilov & Hutter, 2019) (lr=$10^{-3}$, weight decay 0.05) with cosine scheduling, 500-step warmup, and label smoothing of 0.05. We report validation perplexity (the exponentiated mean negative log-likelihood per token; lower is better) and use paired $t$-tests over matched seeds with Cohen's $d$ and 95% confidence intervals. For the main depth result, we use 5 seeds; for the capacity ablation and most communication experiments, 3 seeds; several early screening experiments use 2 seeds, which we note explicitly in Table 1. The additional controls introduced in §6.3 use 1 seed each, except topk_depth (3 seeds); we interpret single-seed results directionally rather than statistically.

For the depth result and capacity ablation (§6.2), three of the four conditions are matched in expert FLOPs per token per layer at $\approx$4M: the standard baseline activates $k$=2 experts of dimension 2048; expert depth

Table 1: Summary of expert communication experiments. $\Delta\%$ is relative change in validation perplexity vs. standard MoE (lower is better). Rows with 2 or 1 seed are screening-level evidence; rows with 3 seeds report $p$-values from paired $t$-tests where shown. No configuration evaluated with 3 seeds achieves a statistically significant improvement; the screening-level rows are directionally consistent, and several variants increase perplexity. Per-seed details are in Appendix A.

| # | Approach | Experts | Timing | Seeds | $\Delta\%$ | Verdict |
|---|----------|---------|--------|-------|-----------|---------|
| 1 | Post-FFN learnable topology | 8 | Post | 2 | $-0.06$ | No improv.[†] |
| 2 | Post-FFN learnable (15 epochs) | 8 | Post | 3 | $+0.46$ | Not sig. ($p$=0.31) |
| 3 | Post-FFN learnable (16 experts) | 16 | Post | 2 | $+0.53$ | No improv.[†] |
| 4 | Post-FFN learnable (32 experts) | 32 | Post | 2 | $-0.34$ | No improv.[†] |
| 5 | Pre-FFN communication | 32 | Pre | 3 | $-0.26$ | Not sig. ($p$=0.63) |
| 6 | Post-FFN (32 experts, high spec.) | 32 | Post | 2 | $+1.26$ | Worse[†] |
| 7 | Pre-FFN + Post-FFN combined | 32 | Both | 2 | $+2.41$ | Worse[†] |
| 8 | Multi-domain (En-De + En-Fr) | 32 | Pre | 1 | $+0.45$ | Worse[†] |
| 9 | Sequential chain (diff. experts) | 32 | Seq. | 3 | $-0.46$ | Not sig. ($p$=0.81) |
| 10 | Topology-governed chain | 32 | Seq. | 3 | $-0.56$ | Not sig. ($p$=0.84) |

[†]Screening-level (1–2 seeds); not formally tested.

activates 1 expert twice through a 2048-dim FFN; and the width control activates 1 expert of dimension 4096. A fourth condition—a single-expert control ($k$=1, dimension 2048, $\approx$2M FLOPs)—uses half the compute budget to isolate the specific contribution of top-$k$=2 ensembling.

## 3 Expert Communication Shows No Clear Gain

**Overview.** We evaluate ten approaches to expert communication across seven axes. Table 1 summarizes all results; no configuration evaluated with 3+ seeds shows a statistically significant improvement over standard independent-expert MoE, and the screening-level results are directionally consistent.

**Parallel communication (rows 1–8).** We implemented a learnable communication topology parameterized as $\sigma(\mathbf{W})$ where $\mathbf{W} \in \mathbb{R}^{N \times N}$ are learned logits and $\sigma$ is the sigmoid function. The $N \times N$ matrix is not the message payload; it only weights expert-to-expert exchange (which pair of experts attends to which). The communicated state is itself $d$-dimensional: each expert's hidden state $E_i(\mathbf{x}) \in \mathbb{R}^d$ is passed through a learned projection $W_{\mathrm{msg}} \in \mathbb{R}^{d \times d}$, aggregated across experts using the topology-derived attention (a graph-attention-style aggregation over the expert graph (Kipf & Welling, 2017; Veličković et al., 2018)), gated per expert by a learned gate (a sigmoid over a linear layer whose $2d$-dimensional input is the concatenation [current state, incoming message], with bias initialized to $-2.0$ so the channel starts suppressed), and added to the original state via a residual connection. The channel is therefore not scalar-per-pair; messages are full $d$-dimensional vectors with a full-rank initial parameterization. This does not rule out richer designs (higher-dimensional messages, cross-attention between experts, learned key/query routing), which we did not test. We evaluated this at multiple expert counts (8, 16, 32), training durations (5, 15 epochs), communication timings (before FFN, after FFN, both), and with multi-domain data (En-De + En-Fr with top-$k$=1 to force expert isolation). At 15 epochs with 8 experts, we also compared the learned topology against two *fixed* alternatives (random and fully-connected all-ones adjacency; 3 seeds each): both are directionally worse than no communication, neither significantly (Appendix A). At 32 experts with high-specialization routing (row 6 of Table 1), post-FFN communication increases perplexity by 1.26%; combining pre- and post-FFN yields the largest degradation at +2.41%. Multi-domain training does not induce domain specialization: expert domain purity remains low with shared BPE tokenization (see Appendix A for the metric and values), so the intended input asymmetry was not realized.

Interestingly, across all parallel communication experiments, models *actively learn to use* the communication mechanisms. Pre-FFN gate biases move from their suppressive initialization toward zero, topology density grows organically, and gradients flow through the message-passing parameters (quantitative details in Appendix A). Yet validation perplexity does not improve. This suggests that diagnostic evidence of mechanism

usage—open gates, growing topology, non-zero gradient norms—is not sufficient evidence that a mechanism is beneficial, a point we believe is methodologically important for evaluating architectural innovations more broadly.

**Sequential composition (rows 9–10).** Motivated by Chain-of-Experts (Wang et al., 2025), we also implemented sequential expert processing where tokens flow through a chain of experts, with each expert seeing the previous expert's output. In the "free" mode (row 9), per-step routing is unconstrained; in "topology-governed" mode (row 10), a learned adjacency matrix constrains allowed expert transitions. Neither achieves significant improvement ($-0.46\%$, $p=0.81$ and $-0.56\%$, $p=0.84$ respectively, both over 3 seeds). The learned topology converges to near-full density (effectively unconstrained; see Appendix A), suggesting the transition structure provides no useful inductive bias in this setting.

## 4 Analysis: A Hypothesis on Mixing Redundancy

The consistency of the negative results across the tested approaches suggests a structural explanation rather than implementation-specific failures. We hypothesize that a key contributing factor is a property of standard MoE with shared inputs: **explicit learned message-passing between parallel experts may duplicate the implicit mixing that the router's weighted output combination already performs**.

Eq. 1 computes $\mathbf{y} = \sum_{i \in \mathcal{S}(\mathbf{x})} g_i(\mathbf{x}) \cdot E_i(\mathbf{x})$. All selected experts in $\mathcal{S}(\mathbf{x})$ receive the same per-token input $\mathbf{x}$ and contribute to $\mathbf{y}$ through gates $g_i$ that are learned end-to-end (Shazeer et al., 2017; Fedus et al., 2022). The router's weighted sum is therefore itself a learned coordination mechanism, operating on exactly the per-token quantities an inter-expert channel would transmit. Consistent with this mixing being genuinely useful, a single-expert dense model with matched FFN FLOPs underperforms the MoE baseline by 6.7% in a single-seed control (§6.3). We do not claim that the router's mixing forecloses a richer parameterization on a priori grounds—a learned channel with its own projection and gate is not strictly contained in the function class of weighted sums. The hypothesis is instead grounded in the empirical pattern: across the parameterizations we test (§3: pre-/post-/both FFN timing; learnable, random, and fully-connected topology; 8, 16, and 32 experts; 5- and 15-epoch training; single- and multi-domain data), no variant improves perplexity, while the gates open from their suppressive initialization and topology density grows organically (Appendix A)—the channel is being used, but not in a way that helps.

**Different experts, same per-token inputs.** We want to be precise about what this hypothesis does *not* say. Across the training distribution, experts are different functions: the router assigns different token sub-distributions to different experts, and they learn distinct transformations. A per-expert input-distribution analysis (Appendix E) quantifies this directly: the normalized pairwise Jensen-Shannon divergence between experts' input-token distributions is 0.89 at layer 0, declining to 0.52 by layer 3. Our claim concerns per-token information flow, not expert specialization: at the moment of forward computation for a single token $\mathbf{x}$, every selected $E_i$ sees the same $\mathbf{x}$, so the messages $\{E_i(\mathbf{x})\}_i$ that an inter-expert channel aggregates are functions of one shared input that already enter the output through the router's weighted sum. This per-token symmetry, not function-level identity across experts, is what we hypothesize makes the parallel-communication channel structurally redundant with the router's mixing in this setting.

**Expert depth has a different computational role.** Same-expert depth breaks the per-token symmetry: pass 2 processes $E(\mathbf{x})$, not $\mathbf{x}$—the input to the expert nonlinearity has changed. By contrast, our parallel-communication variants operate on same-stage expert states that are all derived from the shared token representation. The mechanistic analysis of the depth path supports this distinction: pass 2 acts as a contracting, anti-aligned correction to pass 1 (§7). Sequential composition with a *different* expert, $E_b(E_a(\mathbf{x}))$, also processes a transformed input at its second stage; that variant is null in our experiments (§3, rows 9–10) plausibly for a different reason: the compute is split across two independently optimized functions, and composing through an unrelated expert does not produce representations more useful than those the parallel baseline obtains from soft-ensembling.

**Consistent evidence and limitations.** The hypothesis is consistent with four empirical patterns in this setting: (i) parallel communication is null or harmful across timing, topology, and expert-count combinations (§3); (ii) the tested channel is not scalar-per-pair—the communicated state is $d$-dimensional and passes

---
**Algorithm 1** Expert Depth Forward Pass (per MoE layer)

---
**Require:** Token representations $\mathbf{X} \in \mathbb{R}^{B \times d}$, experts $\{E_1, \ldots, E_N\}$, router $g$, number of passes $T$
1: **logits** $\leftarrow g(\mathbf{X})$ ▷ Router invoked once; logits reused across passes ($B \times N$)
2: $\mathbf{H} \leftarrow \mathbf{X}$ ▷ Current hidden states
3: **for** $t = 1$ to $T$ **do**
4:     $\mathbf{p} \leftarrow \text{softmax}(\textbf{logits})$ ▷ Routing probabilities (same logits each pass)
5:     $\mathbf{idx}, \mathbf{s} \leftarrow \text{top-1}(\mathbf{p})$ ▷ Expert index and score per token
6:     **for** each expert $i = 1, \ldots, N$ **do**
7:         $\mathcal{M}_i \leftarrow \{b : \mathbf{idx}_b = i\}$ ▷ Tokens assigned to expert $i$
8:         **if** $\mathcal{M}_i \neq \emptyset$ **then**
9:             $\mathbf{H}[\mathcal{M}_i] \leftarrow E_i(\mathbf{H}[\mathcal{M}_i]) \cdot \frac{\mathbf{s}[\mathcal{M}_i]}{\text{sg}(\mathbf{s}[\mathcal{M}_i])}$ ▷ Straight-through
10:         **end if**
11:     **end for**
12: **end for**
13: **return** $\mathbf{H}$

---

through a learned $d \times d$ projection and gate (§3)—though we did not test richer designs such as higher-dimensional messages or cross-attention between experts, so channel-design limitations are not ruled out; (iii) depth, the one tested mechanism that changes the per-token input to the expert computation, is the one positive result (§5); (iv) at a doubled FLOP budget where depth is already exploited, adding cross-slot message passing leaves perplexity unchanged in a matched-seed comparison (10.67 vs. 10.66; §6.3). The hypothesis predicts that explicit communication should help most when the communicated quantity cannot be reconstructed from the router's mixing—for instance, when experts receive genuinely asymmetric inputs. Our one attempt to construct such a setting (multi-domain En-De + En-Fr with top-$k$=1; row 8 of Table 1) failed to induce expert specialization under shared BPE (Appendix A), so this prediction was not actually exercised. We emphasize that the hypothesis is consistent with—not validated by—our observations; a rigorous test requires settings where input asymmetry can be independently verified, which we identify as future work.

## 5 Expert Depth: The Same Expert Twice

**Method.** While investigating sequential expert chains, we discovered that applying the **same** router-selected expert twice produces a significant improvement of 7.4% in validation perplexity. Given token $\mathbf{x}$ and router-selected expert $E_a$, standard MoE computes:

$$\mathbf{y}_{\text{parallel}} = g_a \cdot E_a(\mathbf{x}) + g_b \cdot E_b(\mathbf{x}) \quad \text{(two different experts, one pass each)} \tag{2}$$

Expert depth instead applies the same expert twice:

$$\mathbf{y}_{\text{depth}} = E_a\big(E_a(\mathbf{x})\big) \quad \text{(one expert, two passes)} \tag{3}$$

The router selects a single expert via top-1, which processes the token, then processes its own output. Both Eq. 2 and Eq. 3 perform two expert forward passes per token, matching FLOPs. Implementation uses straight-through score estimation—routing scores are multiplied as $s/\text{sg}(s)$ (where sg denotes stop-gradient) to maintain unit forward magnitude while enabling gradient flow through the router. Algorithm 1 gives the pseudocode.

**Results.** Table 2 presents the main result across five random seeds. All five seeds favor expert depth, with a highly significant paired difference ($p = 0.0025$, Cohen's $d = -3.01$).

**The depth benefit grows over training.** Figure 2 shows the epoch-by-epoch trajectory. Expert depth starts behind at epoch 1, reflecting the initial disadvantage of top-1 routing, but crosses over at epoch 2–3 and the gap widens monotonically through epoch 5. This crossover is noteworthy: it indicates that the productive second pass is a *learned* capability, not an automatic architectural benefit—the expert must

Table 2: Expert depth main result (5 seeds, 5 epochs). S_depth applies the same expert twice; all seeds favor it with ~4× lower cross-seed standard deviation. Differences are computed from unrounded per-seed values.

|  | s42 | s123 | s456 | s789 | s1024 | Mean | Std |  |
|---|---|---|---|---|---|---|---|---|
| A_none ($k$=2, 1 pass) | 15.22 | 14.36 | 15.51 | 15.04 | 15.28 | 15.08 | 0.44 | |
| S_depth ($k$=1, 2 pass) | 13.99 | 13.77 | 13.95 | 14.11 | 14.01 | 13.97 | 0.12 | |
| Difference | −1.22 | −0.59 | −1.57 | −0.93 | −1.27 | −1.12 | | **−7.4%** |

Paired $t$-test: $t = -6.73$, $p = 0.0025$, 95% CI $= [-1.58, -0.66]$, Cohen's $d = -3.01$

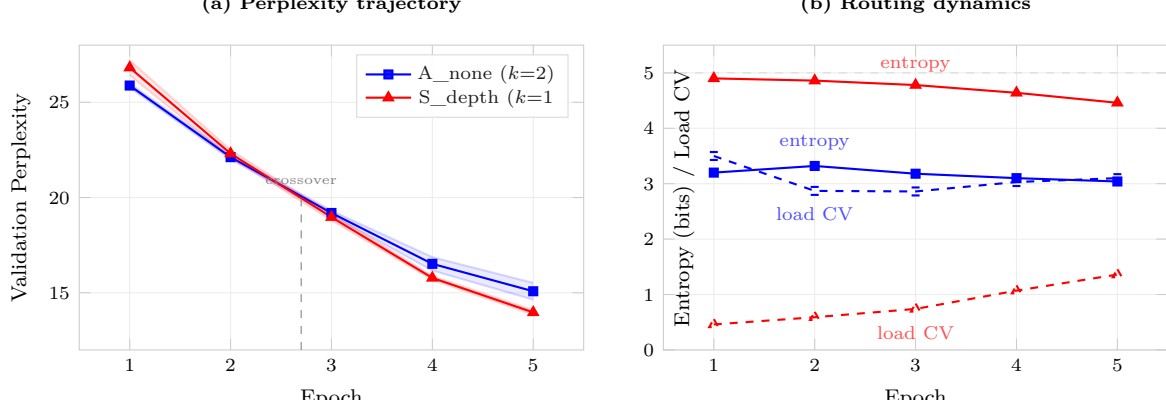

**(a) Perplexity trajectory**  **(b) Routing dynamics**

Figure 2: Expert depth training dynamics (mean ± std of 5 seeds). **(a)** Perplexity: depth starts behind due to top-1 routing but overtakes the baseline as the expert learns to productively process its own output. **(b)** Routing entropy (solid, higher = more uniform) and expert load CV (dashed, lower = more balanced), layer-averaged. Depth maintains near-maximal entropy and 2–3× better load balance.

discover how to process its own output, and this takes several epochs. §6.1 examines how the comparison changes when the baseline is instead given a matched wall-clock budget.

**Improved stability and load balance.** Beyond perplexity, expert depth exhibits qualitatively different training dynamics (Figure 2b). We quantify routing behavior with two diagnostics, defined here since we use them throughout. Let $\bar{p}_i$ denote the fraction of routing assignments (across tokens and top-$k$ slots) that select expert $i$, for $i = 1, \ldots, N$. *Routing entropy* is $H = -\sum_{i=1}^{N} \bar{p}_i \log_2 \bar{p}_i$, measured in bits, with maximum $\log_2 N = 5$ for $N$=32 attained under perfectly uniform expert utilization. *Load CV* is the coefficient of variation $\sigma(\bar{p})/\mu(\bar{p})$ over experts; lower means more balanced load. Under expert depth, the cross-seed standard deviation of final perplexity drops by ~4× (0.44 → 0.12), suggesting a smoother optimization landscape. Load balance also improves substantially: final-epoch load CV is 1.4 under depth vs. 3.1 for the baseline, and routing entropy remains near-maximal throughout training, indicating far more uniform expert utilization. One possible explanation is that top-$k$=2 creates competitive dynamics in which popular experts attract disproportionate load, whereas top-1 selection reduces this effect. Per-epoch dynamics and runtimes are tabulated in Appendix C.

## 6 Compute Analysis, Capacity Ablations, and Controls

### 6.1 FLOP-matched vs. wall-clock-matched comparisons

The answer to "does expert depth help?" depends on how compute is matched; Table 3 summarizes the two comparisons.

Table 3: The depth result under two compute-matching conventions. *FLOP-matched*: both conditions perform two expert forward passes per token per layer and train for the same 5 epochs (5 seeds, paired). *Wall-clock-matched*: A_none trains for 12 epochs so that total time on one H200 approximately equals S_depth's 5 epochs ($12 \times 262 \approx 5 \times 645$ minutes); this comparison is single-seed (42), and the longer-trained baseline performs $\sim 2.4\times$ more total expert FLOPs.

| Comparison | A_none val PPL | S_depth val PPL | Outcome |
|---|---|---|---|
| FLOP-matched, 5 epochs (5 seeds) | 15.08 | **13.97** | S_depth lower by 7.4% ($p$=0.0025) |
| Wall-clock-matched, single seed | **10.46** | 13.99 | A_none lower by 25.2% |

At matched FLOPs and equal training epochs, expert depth is a consistent, statistically significant improvement (§5). It is not, in our implementation, a wall-clock win: the two passes are serialized (and top-1 dispatch further limits batching), so S_depth takes $\sim 2.5\times$ longer per epoch (645 vs. 262 minutes on one H200), and a baseline given approximately the same wall-clock budget trains for $\sim 12$ epochs and reaches lower validation perplexity. Two caveats keep the wall-clock comparison in perspective: it is not FLOP-matched (A_none at 12 epochs performs $\sim 2.4\times$ more total expert FLOPs, which helps explain its lower perplexity), and the penalty reflects serial scheduling in our implementation rather than arithmetic cost—fused multi-pass kernels could narrow the gap, but we have not built them. The precise statement of our positive result is therefore: *same-expert depth is a FLOP-efficiency improvement in this setting, not a demonstrated wall-clock improvement.* A preliminary single-seed BLEU evaluation on newstest2014—in which S_depth scores higher despite the baseline's lower validation perplexity, plausibly because teacher-forced prediction and free-running generation measure different things—is reported with full caveats in Appendix D; the perplexity comparisons are the primary evidence.

## 6.2 Capacity ablation: depth, not width

A natural objection is that expert depth improves simply because it provides more computation per token. To test this, we evaluate four conditions (3 seeds each) that decompose the contributions of routing diversity ($k$), expert width, and iterative depth. An implementation note: the router is invoked exactly once per token per layer in all four conditions. A_single and D_wide use the standard parallel forward path (Eq. 1) with top-$k$=1, applying their single selected expert once. S_depth uses the sequential forward path (Algorithm 1), which takes the same single top-$k$=1 router output and applies the selected expert twice, reusing the router logits across both passes; routing scores are propagated through both passes via straight-through estimation ($s$/sg($s$); Appendix B). Routing is therefore identical between conditions—the same softmax + top-1 selection—and only the post-routing forward path differs: S_depth passes the expert output through the same expert a second time. (The parallel path renormalizes the selected scores, so at top-$k$=1 both paths apply the expert output at unit forward weight.) Table 4 lists the four conditions.

Table 4: Capacity ablation conditions. Expert FLOPs are matched at $\sim 4$M per token per layer across A_none, S_depth, and D_wide. A_single ($\sim 2$M FLOPs) isolates the top-$k$ contribution. The Params column lists expert parameters per layer.

| Condition | Configuration | s42 | s123 | s456 | Mean | Std | Params |
|---|---|---|---|---|---|---|---|
| A_none | $k$=2, dim=2048, 1 pass | 15.22 | 14.36 | 15.51 | 15.03 | 0.60 | 67M |
| A_single | $k$=1, dim=2048, 1 pass | 16.36 | 16.38 | 16.36 | 16.37 | 0.01 | 67M |
| D_wide | $k$=1, dim=4096, 1 pass | 16.38 | 16.40 | 16.34 | 16.37 | 0.03 | 134M |
| S_depth | $k$=1, dim=2048, 2 pass | 13.99 | 13.77 | 13.95 | 13.90 | 0.12 | 67M |

Figure 3 visualizes these results.

**Results.** The ablation reveals three findings. First, the single expert and the width-doubled expert achieve *identical* perplexity (16.37 vs. 16.37, $p$=0.69), despite the latter having $2\times$ the expert parameters (134M vs. 67M per layer) and $2\times$ the per-token FLOPs: doubling expert width provides no benefit. Second, dropping from top-$k$=2 to top-$k$=1 costs +8.9% perplexity (A_none $\rightarrow$ A_single), indicating that soft-ensembling of

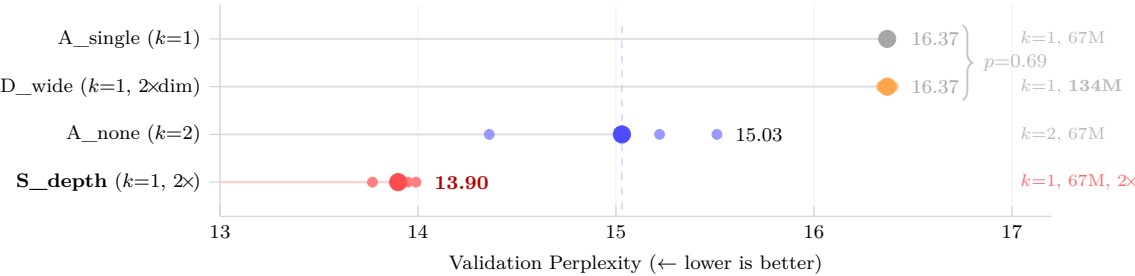

Figure 3: Capacity ablation (3 seeds; large dots are means, small dots are individual seeds). D_wide (134M expert params per layer) matches A_single (67M) exactly despite double the parameters. The observed improvement is attributable to the second pass, not added capacity.

Table 5: Depth × diversity and dense controls, grouped by FFN-FLOP class (FLOPs per token relative to A_none; attention FLOPs scale with layer count). Multi-seed rows report means over seeds; single-seed rows use seed 42. For matched-seed comparisons, the seed-42 values of the multi-seed rows are: A_none 15.22, S_depth 13.99, topk_depth 10.66. Lower val PPL is better.

| Model | Configuration | val PPL | $n$ | FFN FLOPs | Attn FLOPs |
|---|---|---|---|---|---|
| dense_baseline | 1 expert, 4L, FFN 4096, 1 pass | 16.24 | 1 | 1× | 1× |
| A_none | $k$=2, 4L, FFN 2048, 1 pass | 15.08 | 5 | 1× | 1× |
| S_depth | $k$=1, 4L, FFN 2048, 2 passes | 13.97 | 5 | 1× | 1× |
| dense_deep | 1 expert, 8L, FFN 2048, 1 pass | 13.58 | 1 | 1× | 2× |
| dense_16l | 1 expert, 16L, FFN 2048, 1 pass | 11.57 | 1 | 2× | 4× |
| topk_depth | $k$=2, 4L, FFN 2048, 2 passes each | **10.81 ± 0.16** | 3 | 2× | 1× |
| depth_comm | topk_depth + message passing | 10.67 | 1 | 2× | 1× |

two expert outputs contributes substantially to perplexity. Yet expert depth overcomes this disadvantage: despite using top-1 routing, it surpasses the top-$k$=2 baseline by $-7.5\%$ in this 3-seed cohort ($-7.4\%$ over the full five seeds, Table 2). Third, expert depth outperforms both single-pass baselines by $-15.1\%$ ($p = 0.001$). Since all three top-1 conditions share the same routing, the only difference is that expert depth applies the selected expert twice. This 15% improvement is consistent with the second pass through the same expert being the active factor, rather than increased capacity or architectural complexity.

### 6.3 Depth × diversity and dense controls

The ablation above isolates *why* depth helps at matched FLOPs, but leaves two questions open: is the gain specific to MoE, and does depth substitute for routing diversity or combine with it? We ran five additional controls to answer both (Table 5; designs and trajectories in Appendix F). Except for topk_depth (3 seeds), these are single-seed controls we interpret directionally; the dense controls match *activated* per-token FFN FLOPs, not parameter count.

**MoE beats dense at matched FFN FLOPs.** At the matched seed, dense_baseline reaches 16.24 vs. A_none's 15.22—the dense model's perplexity is 6.7% higher, directional evidence that the router's learned mixing is genuinely useful (§4).

**At the $1\times$ FFN-FLOP class, the depth gain is not MoE-specific.** dense_deep—a plain 8-layer dense Transformer with the same total FFN FLOPs as the 4-layer conditions—reaches 13.58, slightly below S_depth's 13.99 at the matched seed ($-2.9\%$), with the caveat that its 8 attention layers double attention FLOPs, so on total compute the comparison is closer or flips. In this shallow regime, added FFN depth helps whether implemented as same-expert iteration inside an MoE layer or as additional dense layers; same-expert depth is best seen as one parameter-efficient way to buy effective depth, not a uniquely MoE-specific mechanism.

**At the** $2\times$ **class, depth and diversity stack—and the combination beats dense depth.** topk_depth applies each of the top-2 selected experts twice and combines the two depth outputs with the router's scores; it reduces to S_depth at $k{=}1$. It reaches $10.81 \pm 0.16$ over 3 seeds—a large jump relative to the $1\times$ conditions ($-22.6\%$ vs. S_depth, $-28.3\%$ vs. A_none; these are not FLOP-matched comparisons, as topk_depth performs $2\times$ the expert FLOPs). The gains from diversity ($k{=}1 \rightarrow k{=}2$, $+8.9\%$ in Table 4) and from depth do not cannibalize each other: depth and diversity are complementary in this setting, not substitutes. The FFN-FLOP-matched dense control at this class, dense_16l (16 layers), reaches 11.57 (topk_depth is 6.6% lower) despite dense_16l spending $4\times$ the attention FLOPs—suggesting, directionally (single seed), that at the $2\times$ operating point routing, expert diversity, and expert depth together allocate compute better than pure dense depth-scaling.

**Communication adds nothing on top of depth and diversity.** depth_comm is identical to topk_depth except that the learned message-passing block of §3 runs between the two depth passes, letting the selected experts' intermediate states exchange information. At the matched seed, depth_comm reaches 10.67 vs. topk_depth's 10.66 ($\Delta = +0.01$, well within topk_depth's cross-seed std of 0.16). This closes a loophole in §3: one could argue the original parallel-communication variants failed because the $1\times$ models were compute-starved. Here, at the $2\times$ class with depth already exploited, explicit communication still adds nothing—consistent with the mixing-redundancy hypothesis (§4).

**Wall-clock.** Unlike S_depth's sequential top-1 chain, topk_depth's two passes run over parallel top-2 dispatch and are comparatively GPU-friendly: the clean seed-42 run averaged $\sim$295 minutes per epoch on one H200 ($\sim$1.13$\times$ A_none's 262, versus S_depth's 645). Some epochs of the other two seeds encountered cluster-slot contention, so we report per-epoch times as indicative only (Appendix F).

## 7 Discussion: Why Does Depth Work?

Each expert $E_i$ is a two-layer FFN: $E_i(\mathbf{x}) = W_2 \cdot \text{GELU}(W_1\mathbf{x} + b_1) + b_2$. Applying it twice creates an effective four-layer transformation with shared weights, structurally analogous to weight-tied iteration in Universal Transformers (Dehghani et al., 2019) and Deep Equilibrium Models (Bai et al., 2019), but applied *within* the sparse expert routing framework rather than across layers. A natural reading of this analogy—and the one we offered in an earlier version of this paper—is that the second pass performs fixed-point-style iterative refinement. We tested this reading directly.

**The second pass is a contracting, anti-aligned correction, not fixed-point iteration.** On a retrained seed-42 checkpoint pair (validation perplexities within 0.3% of the published values), we captured the MoE-layer input $\mathbf{x}$, the pass-1 output $\mathbf{h}_1 = E(\mathbf{x})$, and the pass-2 output $\mathbf{h}_2 = E(\mathbf{h}_1)$ for 102,591 validation target tokens, and examined the pass updates $\Delta_1 = \mathbf{h}_1 - \mathbf{x}$ and $\Delta_2 = \mathbf{h}_2 - \mathbf{h}_1$ (per-layer values in Table 8, Appendix C). Fixed-point convergence would require $\mathbf{h}_2 \approx \mathbf{h}_1$; instead, $\cos(\mathbf{h}_1, \mathbf{h}_2) \approx 0$ at every layer, and at most 0.15% of tokens exceed $\cos > 0.95$. Nor is the expert close to an identity map: pass 1 moves the representation by more than its own magnitude ($\|\Delta_1\|/\|\mathbf{x}\| \approx 1.1$), ruling out a near-identity expert as the trivial explanation. What the second pass actually applies is a *contracting, anti-aligned correction*: $\Delta_2$ points against $\Delta_1$ ($\cos(\Delta_1, \Delta_2) \approx -0.4$) at roughly half its magnitude ($\|\Delta_2\|/\|\Delta_1\| \approx 0.48$), and 99.9% of tokens contract. The two passes also increasingly do different work at later layers: the absolute cosine between the top principal directions of $\Delta_1$ and $\Delta_2$ (computed within each selected expert's token subset) falls monotonically from 0.998 (layer 0) through 0.990 and 0.477 (layers 1–2) to near-orthogonal 0.011 at layer 3. The behavioral difference needs no gating or conditional computation: pass 2 sees a shifted input distribution ($\cos(\mu_\mathbf{x}, \mu_{\mathbf{h}_1}) = 0.07$–0.40 across layers), and a fixed nonlinear function evaluated on a shifted distribution implements a different transformation. These analyses are single-seed (42) and descriptive rather than causal (§9).

**The learned correction develops over training.** The training trajectory (Figure 2) shows how this behavior emerges. In early training, the expert is optimized for single-pass processing, and applying it twice initially degrades perplexity; over training, the expert adapts to process both raw inputs and its own outputs. The crossover appears gradual rather than sudden, suggesting continuous adaptation of the expert weights rather than a sharp phase transition. The contraction ratio also yields a mechanistic prediction for more passes: modeling pass-$t$ updates as geometric attenuation with ratio $r \approx 0.48$ puts a 4-pass model at $\sim$95%

of the infinite-pass limit, predicting rapid diminishing returns beyond $T=2$. This is a prediction from the 2-pass checkpoint, not a measurement; we have not trained 3- or 4-pass models (§9).

**Depth changes what routing is for—but not by finding "the right expert."** Standard top-$k$ MoE can be viewed as an *ensemble*: it combines $k$ different expert opinions about each token. We initially expected expert depth to operate as an iterative solver that selects the single most relevant expert and refines with it. The routing statistics contradict the "stable specialist" half of that picture: under depth, routing becomes *more* diffuse, not more committed. Final-epoch routing entropy reaches 89% of its maximum (vs. 61% for the baseline; load CV 1.36 vs. 3.10), and at validation time the same vocabulary type spreads across many experts: the largest per-token-id expert share at layer 0 is 0.23 under depth vs. 0.64 for the baseline (per-layer values in Appendix C). A consistent reading is that whichever expert the learned, context-dependent router selects for a given occurrence, the second pass applies an attenuated, anti-aligned correction to that expert's first-pass output—the router does not need to commit a token type to one specialist for the mechanism to operate. Token-level attribution supports this corrective picture: only 48.7% of tokens individually improve under depth; gains concentrate at mid-sentence positions (2–5× the improvement at sentence edges), where autoregressive context is richest; and the mean improvement on high-frequency tokens comes from a compressed loss tail (fewer large errors) rather than uniform gains (full breakdowns in Appendix C). We therefore describe expert depth as *learned context-dependent routing followed by an attenuated corrective second pass*, and we soften our earlier framing that the router's value lies in selecting "the right expert": the depth gain does not rely on stable token-to-expert specialization in our experiments.

# 8  Related Work

**Mixture-of-Experts.** MoE dates to Jacobs et al. (1991) and Jordan & Jacobs (1994), with the modern sparse gating formulation by Shazeer et al. (2017). Switch Transformer (Fedus et al., 2022) simplified routing to top-1; GShard (Lepikhin et al., 2021) demonstrated distributed MoE training. Expert Choice routing (Zhou et al., 2022) inverts the assignment: experts select tokens rather than tokens selecting experts. BASE Layers (Lewis et al., 2021) formulate routing as balanced assignment via optimal transport, and Hash Layers (Roller et al., 2021) replace the learned router with fixed hashing—both modify *how* tokens reach experts rather than whether experts interact. Recent large-scale deployments include Mixtral (Jiang et al., 2024) and DeepSeekMoE (Dai et al., 2024); the latter introduced shared experts alongside routed experts as a form of implicit coordination, an approach extended by DeepSeek-V3 (DeepSeek-AI, 2024). ST-MoE (Zoph et al., 2022) studied stability and transferability; GLaM (Du et al., 2022) demonstrated efficient scaling; and sparse upcycling (Komatsuzaki et al., 2023) converts dense models to MoE. All of these treat routed experts as independent processors—our negative results on explicit inter-expert communication are consistent with this design choice. Scaling laws for MoE (Clark et al., 2022; Kaplan et al., 2020; Hoffmann et al., 2022) and differentiable expert selection (Hazimeh et al., 2021) have been studied but do not address inter-expert interaction.

**Expert communication and interaction.** Chain-of-Experts (Wang et al., 2025) proposes sequential expert composition with different experts at each step, reporting gains on math reasoning with DeepSeek-V2-Lite. Our sequential-different-expert condition (row 9 in Table 1) replicates this approach and finds no significant gain on translation, suggesting task dependence. Our key finding—that same-expert iteration outperforms different-expert chains—is novel relative to CoE, which explicitly routes to different experts per step. Soft MoE (Puigcerver et al., 2024) replaces discrete routing with continuous token-expert mixing. V-MoE (Riquelme et al., 2021) scales vision transformers with MoE layers. OLMoE (Muennighoff et al., 2024) provides open MoE language models. SwitchHead (Csordás et al., 2024) applies MoE to attention heads. Chen et al. (2022) analyze the theoretical properties of MoE layers. None of these works proposes or evaluates iterative expert application. Our learned communication topology uses graph-based parameterization related to GCNs (Kipf & Welling, 2017) and GATs (Veličković et al., 2018), applied to expert adjacency.

**Weight-tied iterative processing.** Universal Transformers (Dehghani et al., 2019) share weights across transformer layers, applying the same transformation iteratively with halting mechanisms. Deep Equilibrium Models (Bai et al., 2019; 2020) solve for fixed-point representations, finding that implicit-depth models can match explicit-depth ones. Unlike DEQ-style convergence toward a fixed point, the mechanism we observe in

expert depth is a single attenuated, anti-aligned correction (§7). Adaptive Computation Time (Graves, 2016) learns to apply variable numbers of iterations per input. ALBERT (Lan et al., 2020) shares parameters across transformer layers to reduce model size. Our expert depth applies weight-tied iteration *within* sparse expert routing—the router selects *which* expert iterates—and our analysis characterizes the result as a corrective second pass rather than refinement toward a fixed point (§7).

**Negative results and reproducibility.** Systematic negative findings prevent wasted research effort: Dacrema et al. (2019) showed that many deep-learning recommendation models fail to beat well-tuned baselines, Lipton & Steinhardt (2019) identified failure to beat baselines among troubling trends in ML scholarship, Musgrave et al. (2020) found similar issues in metric learning, and Bouthillier et al. (2021) emphasized accounting for benchmark variance—a concern we address through multi-seed evaluation. Our study contributes a broad empirical look at expert communication in MoE on a standard translation benchmark, while being explicit that several configurations are supported only by screening-level evidence.

## 9 Limitations and Future Work

Our models are modest in size compared to current state-of-the-art MoE systems. All experiments use WMT14 En-De translation with a $\sim$290M-parameter decoder-only model ($\sim$40M active per token) and 32 experts per layer; every claim in this paper is scoped to this single benchmark and scale, and we do not claim transfer to other tasks or model sizes. We train without an auxiliary load-balancing loss (the baseline's imbalanced routing is consistent with that choice); repeating the comparison with one is a natural unrun control. Wang et al. (2025) report gains from different-expert composition on math reasoning, suggesting task dependence. Scaling to 1B+ parameters with 256+ experts (Lepikhin et al., 2021; Fedus et al., 2022; Rajbhandari et al., 2022) is an important direction. Several controls in §6.3 and the mechanism analyses of §7 are single-seed (42) and descriptive rather than causal; multi-seed replication would strengthen both. We evaluate only $T=2$ sequential passes; the contraction analysis predicts diminishing returns by 3–4 passes (§7), but we have not measured this directly. The depth $\times$ diversity ablation shows the two axes combine productively in this setting (§6.3); the natural remaining control at the 4-call FLOP class is top-$k=4$ with a single pass, which we have not run. On the communication side, we have not exhaustively ablated the channel design (message dimensionality, projection rank, cross-attention between experts), so the mixing-redundancy hypothesis is supported only for the parameterizations we tested. S_depth is $\sim$2.5$\times$ slower per epoch than the baseline due to serial scheduling of the two passes; fused kernels could reduce this overhead, and §6.1 quantifies the resulting wall-clock trade-off. BLEU on newstest2014 (Appendix D) is low in absolute terms for both conditions due to the decoder-only prefix-LM architecture; encoder-decoder or language-modeling evaluations would give a more complete picture. Adaptive computation (Graves, 2016), where the router learns how many iterations each token needs, remains a natural extension.

## 10 Conclusion

This work began with a natural hypothesis—that expert communication should improve MoE performance—and arrived at a more nuanced conclusion. Across ten approaches and seven experimental axes on WMT14 En-De, we find no statistically significant gain among the communication configurations tested with 3+ seeds, six screening experiments are directionally consistent, and communication remains null even when added to our strongest depth-plus-routing configuration at a doubled FLOP budget. Applying the same expert twice, by contrast, yields a significant FLOP-matched improvement, attributable to the second pass rather than added width and characterized mechanistically as a contracting, anti-aligned correction rather than fixed-point refinement. Dense controls temper the interpretation: a FLOP-matched deeper dense model achieves a comparable gain (depth is not MoE-specific at this scale), and a wall-clock-matched baseline trained longer reaches lower perplexity (the result is a FLOP-efficiency finding). The strongest configuration combines routing, expert diversity, and expert depth, outperforming an FFN-FLOP-matched dense model at the doubled budget. On WMT14 En-De at this scale, our results therefore point toward giving selected experts deeper processing—alone or stacked with routing diversity—and away from explicit inter-expert communication, whose tested forms appear redundant with the router's mixing; whether this holds beyond translation, at larger scales, or where experts hold genuinely asymmetric inputs remains open.

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

## A   Detailed Negative Results

Table 6 provides per-seed validation perplexity for three communication experiments evaluated with 3 seeds (rows 5, 9, 10 in Table 1). For all remaining rows, we report per-seed values inline below, organized by seed count.

Table 6: Per-seed validation perplexity for experiments with 3 seeds.

| Experiment | Condition | s42 | s123 | s456 | Mean |
|---|---|---|---|---|---|
| Pre-FFN (32 exp) | A_none | 15.16 | 15.01 | 14.95 | 15.04 |
| | E_prefn | 14.98 | 15.05 | 14.98 | 15.00 |
| Sequential chain | A_none | 15.22 | 14.36 | 15.51 | 15.03 |
| | S_free | 14.83 | 14.80 | 15.26 | 14.96 |
| Topology chain | A_none | 15.22 | 14.36 | 15.51 | 15.03 |
| | S_topology | 14.95 | 14.98 | 14.90 | 14.94 |

**3-seed results at 15 epochs (8 experts): learned vs. fixed topologies.** This experiment compared the learned topology against two fixed alternatives under an otherwise matched configuration (row 2 of Table 1 reports the learnable condition). A_none: s42=15.43, s123=15.35, s456=15.39 (mean 15.39). Learnable topology (C_learn): s42=15.54, s123=15.32, s456=15.52 (mean 15.46); $\Delta = +0.46\%$, paired $t$-test $p = 0.31$. Fixed fully-connected topology (all-ones adjacency): s42=15.46, s123=15.52, s456=15.57 (mean 15.52); $\Delta = +0.84\%$, $p = 0.11$. Fixed random topology: s42=15.75, s123=15.31, s456=15.70 (mean 15.59); $\Delta = +1.29\%$, $p = 0.24$. Neither the learned nor the fixed topologies improve over no communication, extended training does not create separation, and both fixed variants are directionally worse than the learned one.

**2-seed screening results.** Post-FFN learnable topology (8 experts, 5 epochs): A_none s42=15.97, s123=15.85 (mean 15.91); C_learn s42=16.00, s123=15.81 (mean 15.90); $\Delta = -0.06\%$. Post-FFN (16 experts): A_none s42=15.56, s123=15.64 (mean 15.60); C_learn s42=15.73, s123=15.64 (mean 15.68); $\Delta = +0.53\%$. Post-FFN (32 experts): A_none s42=14.85, s123=15.63 (mean 15.24); C_learn s42=15.24, s123=15.14 (mean 15.19); $\Delta = -0.34\%$. Post-FFN at 32 experts with high specialization (2 seeds): C_postfn

s42=15.12, s123=15.43 (mean 15.28) vs. A_none s42=15.16, s123=15.01 (mean 15.09); $\Delta = +1.26\%$. Pre-FFN + Post-FFN combined (2 seeds): F_both s42=15.39, s123=15.51 (mean 15.45) vs. A_none s42=15.16, s123=15.01 (mean 15.09); $\Delta = +2.41\%$.

**1-seed screening result.** In the multi-domain setting (En-De + En-Fr, top-$k$=1, seed 42), A_none achieves 22.39 and E_prefn achieves 22.49, for $\Delta = +0.45\%$. This is a single-seed screening result and should be interpreted accordingly.

**Mechanism usage diagnostics.** Across pre-FFN experiments, gate biases moved from their suppressive initialization of $-2.0$ to near-zero values ($-0.18$ to $+0.06$ across layers and runs at epoch 5), indicating the model actively opened the communication channel. Topology density grew organically from 3.1% (identity initialization) to 15–26% in the first three layers (6–10% in the final layer). In the topology-governed sequential experiments, the learned transition graph converged to 97–100% density across all layers and seeds, effectively eliminating the structural constraint.

**Domain specialization.** In the multi-domain experiment (En-De + En-Fr, top-$k$=1), we measure per-expert domain purity as the maximum domain share of the tokens routed to an expert (1.0 = fully domain-specialized; 0.5 = domain-agnostic for two balanced domains), averaged over all 32 experts with unused experts contributing zero. Mean purity was 0.13 (A_none) and 0.20 (E_prefn) at epoch 5: the experts that received tokens sat near the domain-agnostic level, and many experts received few tokens. Experts assigned tokens from both language pairs with near-equal frequency, indicating that the shared BPE tokenizer produces sufficiently similar subword distributions across languages to prevent domain-specific routing.

## B  Implementation Details

**Model architecture.** The model is a Transformer decoder with causal self-attention. Each of the 4 layers contains multi-head self-attention (8 heads, dimension 64 per head) followed by the MoE-FFN sublayer. The MoE-FFN contains 32 expert modules, each a 2-layer FFN: Linear($512 \rightarrow 2048$) $\rightarrow$ GELU $\rightarrow$ Linear($2048 \rightarrow 512$), with layer normalization applied before the MoE sublayer (pre-norm architecture). The output projection and a residual connection follow the MoE output. Total parameters: $\sim$290M, of which $\sim$268M are expert weights (32 experts $\times$ 4 layers $\times$ $\sim$2.1M); with top-$k$=2, $\sim$40M parameters are active for a given token.

**Training hyperparameters.**

| Hyperparameter | Value |
|---|---|
| Optimizer | AdamW ($\beta_1$=0.9, $\beta_2$=0.999) |
| Learning rate | $10^{-3}$ |
| Weight decay | 0.05 |
| LR schedule | Cosine decay (min ratio 0.1) |
| Warmup steps | 500 |
| Batch size | 64 |
| Epochs | 5 |
| Max sequence length | 128 tokens |
| Label smoothing | 0.05 |
| Gradient clipping | 1.0 |
| Dropout | 0.1 |
| Tokenizer | Shared BPE (SentencePiece) |
| Training data | 4.3M pairs (WMT14 En-De) |
| Validation data | 201K pairs (news-commentary) |

**Expert depth implementation.** In depth mode, the sequential forward pass (Algorithm 1 in the main text) reuses the same router logits at both steps. Since the same expert is selected at both steps, the routing is deterministic: the top-1 expert at step 0 is guaranteed to be reselected at step 1. Gradient flow through the routing scores is maintained via straight-through estimation: scores are multiplied as $s/\text{stopgrad}(s)$, which

has unit forward value but allows gradients to propagate to the router parameters. This is implemented in approximately 30 lines of PyTorch within the `ExpertGraph.forward_sequential()` method.

**FLOP computation.** Each expert FFN performs two matrix multiplications: in_proj $\in \mathbb{R}^{512 \times 2048}$ and out_proj $\in \mathbb{R}^{2048 \times 512}$, for a total of $2 \times 512 \times 2048 \approx 2.1M$ multiply-accumulate operations per expert activation. With top-$k$=2, the baseline uses $\approx$4.2M FLOPs per token per layer. Expert depth (2 passes through one expert) uses the same $\approx$4.2M FLOPs. The width-doubled control (dim=4096) uses $2 \times 512 \times 4096 \approx 4.2M$ FLOPs in a single pass.

## C  Additional Depth Diagnostics

**Per-epoch training dynamics.** Table 7 reports per-epoch validation perplexity, training perplexity, routing entropy, and expert load CV for both conditions, averaged across 5 seeds.

Table 7: Per-epoch training dynamics (mean of 5 seeds, layer-averaged). Routing entropy is in bits (max = $\log_2 32 = 5.0$). Load CV is the coefficient of variation of expert token counts (lower = more balanced).

| Epoch | Condition | Val PPL | Train PPL | Entropy | Load CV |
|---|---|---|---|---|---|
| 1 | A_none | $25.88 \pm 0.11$ | 59.38 | 3.20 | 3.50 |
|   | S_depth | $26.82 \pm 0.39$ | 67.54 | 4.90 | 0.46 |
| 2 | A_none | $22.12 \pm 0.11$ | 37.38 | 3.32 | 2.87 |
|   | S_depth | $22.31 \pm 0.19$ | 38.34 | 4.86 | 0.59 |
| 3 | A_none | $19.19 \pm 0.12$ | 32.48 | 3.18 | 2.86 |
|   | S_depth | $18.97 \pm 0.16$ | 33.23 | 4.78 | 0.74 |
| 4 | A_none | $16.52 \pm 0.35$ | 27.52 | 3.10 | 3.03 |
|   | S_depth | $15.78 \pm 0.10$ | 28.27 | 4.64 | 1.07 |
| 5 | A_none | $15.08 \pm 0.44$ | 23.24 | 3.04 | 3.10 |
|   | S_depth | $13.97 \pm 0.12$ | 24.04 | 4.46 | 1.36 |

At epoch 5, normalized routing entropy $H/\log_2 N$ is 0.61 for A_none vs. 0.89 for S_depth.

**Per-layer geometry of the two depth passes (seed 42).** Table 8 gives the per-layer values behind the contracting anti-aligned correction characterized in §7.

Table 8: Per-layer geometry of the two depth passes (seed 42; 102,591 validation target tokens). $\Delta_1 = \mathbf{h}_1 - \mathbf{x}$, $\Delta_2 = \mathbf{h}_2 - \mathbf{h}_1$. The pattern—near-orthogonal $\mathbf{h}_1, \mathbf{h}_2$; anti-aligned updates at half magnitude; pass-1 update larger than the input—characterizes a contracting corrective step, not convergence toward a fixed point.

| Layer | $\cos(\mathbf{h}_1, \mathbf{h}_2)$ | $\cos(\Delta_1, \Delta_2)$ | $\|\Delta_2\|/\|\Delta_1\|$ | $\|\Delta_1\|/\|\mathbf{x}\|$ |
|---|---|---|---|---|
| 0 | 0.006 | $-0.398$ | 0.460 | 1.163 |
| 1 | 0.006 | $-0.451$ | 0.492 | 1.176 |
| 2 | 0.019 | $-0.372$ | 0.505 | 1.095 |
| 3 | 0.028 | $-0.314$ | 0.481 | 1.070 |

**Step overlap verification.** In depth mode, step overlap (fraction of tokens selecting the same expert at both sequential steps) is exactly 1.0 across all 5 seeds, all 4 layers, and all 5 epochs, confirming that the same expert is applied twice as intended.

**Token-level attribution (seed 42).** Scoring 208,502 validation target tokens against both checkpoints with unsmoothed cross-entropy (raw perplexity 7.894 for A_none vs. 7.379 for S_depth; lower than the label-smoothed training-time values, but agreeing in direction), S_depth reduces mean per-token NLL by 0.067 nats. Table 9 breaks the improvement down by target-relative position quartile and by log-frequency quintile. The mean improvement concentrates mid-sentence and, by frequency, rises toward common tokens even as the fraction of individually improved tokens falls—because the gain comes from tail shape: common tokens

lose their large-loss tail (1st percentile $-1.96$ vs. $-3.84$ for rare tokens) rather than improving uniformly. Median per-token change is $\approx 0$ in every quintile, and only 48.7% of tokens individually improve.

Table 9: Per-token NLL improvement of S_depth over A_none (seed 42; positive = improvement, in nats). Left: by target-relative position quartile. Right: by log-frequency quintile, with the fraction of tokens individually improving and the 1st/99th percentiles of the per-token change.

| Position | mean $\Delta$ | Frequency | mean $\Delta$ | frac. improving | p1 | p99 |
|---|---|---|---|---|---|---|
| Q0 (start) | +0.02 | Q0 (rare) | +0.07 | 51.1% | $-3.84$ | +3.81 |
| Q1 | +0.10 | Q1 | +0.05 | 50.1% | $-3.67$ | +3.77 |
| Q2 | +0.10 | Q2 | +0.05 | 47.5% | $-3.42$ | +3.77 |
| Q3 (end) | +0.05 | Q3 | +0.08 | 48.6% | $-2.78$ | +3.17 |
| | | Q4 (common) | +0.09 | 46.5% | $-1.96$ | +2.54 |

**Per-token-id routing diffuseness (seed 42).** For the 3,385 vocabulary types with at least 5 validation occurrences, Table 10 reports the largest fraction of routing mass any single expert receives across a type's occurrences (max-share) and the entropy of that per-type expert distribution, both weighted by occurrence count. Each individual occurrence is still routed top-1; the diffuseness measured here is across the occurrences of a vocabulary type.

Table 10: Per-token-id routing statistics at validation time (seed 42). S_depth spreads occurrences of the same vocabulary type across more experts in layers 0–2; layer 3 inverts the pattern, consistent with stronger final-layer specialization.

| Layer | A_none | | S_depth | |
| | max-share | entropy (bits) | max-share | entropy (bits) |
|---|---|---|---|---|
| 0 | 0.64 | 1.41 | 0.23 | 3.57 |
| 1 | 0.55 | 1.78 | 0.28 | 3.33 |
| 2 | 0.55 | 1.95 | 0.35 | 2.96 |
| 3 | 0.50 | 1.88 | 0.56 | 2.10 |

**Runtime.** A_none averages 262 minutes per epoch; S_depth averages 645 minutes per epoch (2.5× the baseline's per-epoch time) on a single H200 GPU. The overhead is due to the two expert passes, which our implementation serializes. D_wide averages 221 minutes per epoch; A_single averages 220 minutes per epoch.

## D  Preliminary BLEU Evaluation

As a preliminary assessment of downstream generation quality, we evaluated BLEU (Papineni et al., 2002) on the WMT14 newstest2014 test set (2,737 sentences) using greedy decoding with repetition penalty 1.2 and no-repeat 3-gram blocking (single seed; Table 11).

Table 11: Preliminary BLEU on WMT14 newstest2014 (single seed, greedy decoding with repetition penalty 1.2 and no-repeat 3-gram blocking). A_none is trained for 12 epochs to approximately match S_depth's total wall-clock compute (§6.1). Single-run results without variance estimates; interpret with caution.

| Condition | Epochs | Val PPL | BLEU | Length Ratio |
|---|---|---|---|---|
| A_none ($k=2$, wall-clock-matched) | 12 | 10.46 | 4.9 | 1.11 |
| S_depth ($k=1$, 2 pass) | 5 | 13.99 | **11.8** | 1.07 |

Expert depth achieves higher BLEU (11.8 vs. 4.9) despite the wall-clock-matched baseline reaching lower validation perplexity with more training. We deliberately treat this as a side observation rather than evidence, for three reasons: the comparison is single-run with non-standard decoding heuristics; absolute BLEU is low

for both conditions, reflecting the decoder-only prefix-LM architecture rather than a translation-optimized encoder-decoder; and the two metrics measure different things—perplexity is teacher-forced and token-level, while BLEU scores free-running generation in which errors compound autoregressively. The divergence is nonetheless intriguing. A model that predicts held-out text better after longer training (A_none at 12 epochs, evaluated on the news-commentary validation set) need not generate better on a different-domain test set, and the mechanism findings of §7—improvements concentrated at mid-sentence positions and a compressed loss tail on frequent tokens—are consistent with depth helping most where autoregressive compounding hurts greedy decoding, though those teacher-forced analyses do not directly measure generation. We also cannot rule out single-seed noise or decoding-hyperparameter sensitivity. Resolving the divergence would require multi-seed BLEU with standard decoding (e.g., beam search) and stronger architectures, which we leave as future work. The primary evidence for expert depth (validation perplexity, 5 seeds, $p=0.0025$) does not depend on this BLEU comparison.

## E  Per-Expert Input Distributions

To quantify how different the experts are on the input distribution—as opposed to within a single token's forward computation (§4)—we computed, for the A_none baseline (seed 42), the distribution of input tokens routed to each of the 32 experts in each layer over the full validation set (201,288 held-out sequences). For every pair of experts within a layer we computed the Jensen-Shannon divergence between their input-token distributions, normalized by $\ln 2$ so that 1 indicates fully disjoint token distributions and 0 identical ones. Table 12 reports per-layer summaries.

Table 12: Per-expert input-token distribution analysis (A_none, seed 42, full validation set). Pairwise JSD is normalized by $\ln 2$ (1 = disjoint distributions, 0 = identical). Experts receive markedly distinct token distributions in early layers and converge toward overlap by the last layer.

| Layer | Load CV | Mean pairwise JSD | Min | Max |
|---|---|---|---|---|
| 0 | 1.40 | 0.89 | 0.03 | 1.00 |
| 1 | 1.57 | 0.85 | 0.03 | 0.99 |
| 2 | 1.90 | 0.74 | 0.09 | 0.95 |
| 3 | 1.71 | 0.52 | 0.01 | 0.98 |

Qualitatively, layer-0 experts specialize on recognizable token classes: one expert's most frequent inputs are German articles and pronouns (*die, sich, das, eine*), another's German conjunctions and negation (*und, nicht, sie*), another's German prepositions (*in, von, auf, für, mit*), another's German verb forms (*ist, werden, sind, hat, wird*), another's English function words (*the, to, and, a*), and others concentrate on punctuation and sequence boundaries. By layer 3, many experts share the same high-frequency top tokens (commas, periods, articles) and differ mainly in the tails of their distributions, consistent with the declining mean JSD. Load CV in Table 12 is computed from validation-set routing counts for a single seed and therefore differs from the training-time, layer-averaged load CV of Table 7. Experts are therefore markedly different functions *across* tokens, especially in early layers. This does not contradict the mixing-redundancy hypothesis of §4, which concerns per-token information flow: at the moment a single token is processed, all selected experts still receive the same input.

## F  Additional Controls: Designs and Trajectories

This appendix details the five controls of §6.3, all trained with the setup of §2 (full 4.3M-pair WMT14 En-De, 5 epochs, seed 42 unless noted).

**dense_baseline** (1 expert, 4 layers, FFN 4096): replaces each MoE layer with a single FFN whose hidden dimension is doubled so that per-token FFN FLOPs match A_none's two active 2048-dim experts. Tests whether MoE routing + soft-ensembling beats an equal-FLOP dense FFN.

**dense_deep** (1 expert, 8 layers, FFN 2048): a plain 8-layer dense Transformer; total FFN FLOPs match the 4-layer conditions, attention FLOPs double. Tests whether the depth gain is MoE-specific.

**dense_16l** (1 expert, 16 layers, FFN 2048): the dense control at the $2\times$ FFN-FLOP class; attention FLOPs are $4\times$. Tests whether pure dense depth-scaling matches depth $\times$ diversity at equal FFN FLOPs.

**topk_depth** ($k=2$, 4 layers, depth 2; seeds 42/43/44): the router selects the top-2 experts as in A_none; each selected expert is applied twice sequentially, and the two depth outputs are combined via the router's scores. Reduces to S_depth at $k=1$. Four expert calls per token: $2\times$ the expert FLOPs of A_none and S_depth.

**depth_comm** (topk_depth + message passing): identical to topk_depth, with the learned message-passing block of §3 exchanging information between the two selected experts' states between depth passes. Matched FLOP class; isolates the marginal value of communication on top of depth and diversity.

Table 13 gives the per-epoch validation-perplexity trajectories.

Table 13: Per-epoch validation perplexity for the additional controls. topk_depth reports mean over 3 seeds (per-seed epoch-5 values: 10.66 / 10.98 / 10.78); other rows are seed 42.

| Model | Epoch 1 | Epoch 2 | Epoch 3 | Epoch 4 | Epoch 5 |
|---|---|---|---|---|---|
| dense_baseline | 26.36 | 22.76 | 20.01 | 17.56 | 16.24 |
| dense_deep | 26.44 | 20.37 | 17.15 | 14.64 | 13.58 |
| dense_16l | 27.74 | 17.81 | 14.74 | 12.56 | 11.57 |
| topk_depth (3-seed mean) | 19.71 | 16.25 | 13.88 | 11.81 | 10.81 |
| depth_comm | 19.62 | 16.15 | 13.82 | 11.68 | 10.67 |

**Wall-clock.** Approximate first-epoch times on one H200: dense_baseline $\sim$118 minutes, dense_deep $\sim$185, dense_16l $\sim$400, topk_depth $\sim$295–310, depth_comm $\sim$340. Later epochs of some runs are not cleanly measurable because jobs were chained across scheduler walltime limits, and two topk_depth seeds encountered cluster-slot contention; we therefore treat per-epoch times as indicative. The qualitative ordering is robust: parallel top-2 dispatch with depth (topk_depth, $\sim$1.1–1.2$\times$ A_none's 262 minutes) is far cheaper in wall-clock terms than the sequential top-1 chain (S_depth, 645 minutes), and depth_comm's message passing adds $\sim$10% overhead over topk_depth.

