# OpenReview forum: "Same-Expert Iteration Improves a Translation MoE Where Expert Communication Does Not"
_TMLR — Under review for TMLR_

### Review · Reviewer_UZen · 2026-05-06

**Summary Of Contributions:**

The paper systematically evaluates expert communication mechanisms in Mixture-of-Experts (MoE) models for neural machine translation. The authors evaluate ten expert-communication variants across several design axes, including pre or post-FFN communication, learned topologies, sequential expert chains, expert count, training duration, and multi-domain data. They find no clear evidence that communication improves over a standard MoE baseline. The paper's main positive result is expert depth, where the same router-selected expert is applied twice sequentially. This approach achieves substantially lower validation perplexity than the standard MoE baseline, with consistent gains across five seeds. The authors further include a capacity ablation suggesting that the benefit comes from iterative same-expert refinement rather than simply increased width or compute.


Strengths:

1.) The paper addresses a timely and important question in MoE architecture design, whether expert communication is actually useful.

2.) The experimental comparison is diverse, covering multiple communication mechanisms rather than evaluating only one proposed variant.

3.) The negative results are nicely framed as no clear evidence rather than overclaiming evidence of absence, including diagnostic evidence that models actively use communication mechanisms without benefit, methodologically important for the field.

4.) The expert-depth result is interesting and potentially useful, especially because the capacity ablation suggests the gain is not explained by merely doubling expert width. Also, the four-way capacity ablation (standard MoE vs. single expert vs. width-doubled vs. depth) shows that doubling expert width provides no benefit, while the second pass through the same expert accounts for the observed gain, suggesting iterative refinement rather than increased capacity is responsible.

5.) The paper proposes that information asymmetry may be crucial-parallel experts lack genuine information differences, making communication redundant with what the routing layer already provides via soft-ensembling.

6.) I like the transparency of the authors, where they discuss limitations, including small model scale, limited seed counts for some communication variants, low BLEU scores, and the preliminary nature of downstream evaluation.



Weaknesses:

1.) The study uses a modest decoder model (only 93M parameters) on a single translation task (WMT14 En-De). It's unclear whether findings generalize to larger models, other modalities such as vision, speech, or other tasks, like reasoning and classification.

2.) The reported BLEU scores (4.9-11.8) seem to be low for modern translation systems, suggesting the decoder-only architecture is not properly suited for translation. The discrepancy between perplexity and BLEU (lower perplexity but also lower BLEU for compute-matched baseline) raises questions about the metric's reliability in this setup.

3.) They should have explained computational overhead in detail, like expert depth requires ~2.5x longer training time per epoch due to sequential processing, which is acknowledged but not deeply explored.

4.) While the paper shows depth works and attributes it to iterative refinement, there is no mechanistic analysis, such as fixed-point convergence, representation analysis, and attention patterns to support this claim.

5.) Chain-of-Experts (Wang et al., 2025) showed gains on math reasoning but not on translation. This suggests expert communication may be task-dependent; it would have been better if the paper had explored this boundary condition.

6.) All main results are on a modest decoder-only translation model and one primary benchmark. It would have been better if the authors had attempted multiple benchmarks to show the generalizability and efficacy of their proposed approach.

**Audience:**

Yes

**Audience Explanation:**

Yes. The paper would likely interest researchers working on MoE models, sparse routing, efficient transformer architectures, expert specialization, and negative-result methodology. The finding that explicit expert communication fails to improve performance in this setting is useful because many MoE extensions assume that expert interaction should help. The same-expert depth result is also interesting because it suggests a simple alternative using sparse routing to choose the right expert and then applying that expert more deeply. This could inspire further work on iterative experts, adaptive computation, and depth-vs-diversity tradeoffs in MoE architectures.

**Broader Impact Concerns:**

I do not see major ethical concerns requiring substantial additional broader-impact discussion. The work is primarily an architectural study of MoE models on machine translation.

**Claims And Evidence:**

Yes

**Claims Explanation:**

Not all results are clear and convincing.

Like, for the negative communication results, the evidence is clear and appropriately convincing. The paper reports per-seed validation perplexity, p-values, and explicitly notes when results are screening-level. The diagnostic evidence that models actively use communication channels (gate biases moving from -2.0 to near-zero, topology density growing) is particularly convincing that the negative results are not due to implementation failures. Also, the claim is properly framed as "absence of clear evidence" rather than "evidence of absence," which is appropriate given the statistical power limitations. Additionally, for expert depth improvements, the main result (7.4% lower perplexity, p=0.0025, 5 seeds) is statistically robust. The ablation showing width-doubled experts perform identically to single experts (16.37 vs. 16.37, p=0.69) convincingly rules out the capacity explanation. The training dynamics in Fig. 2 provide clear evidence that the improvement emerges gradually as the expert adapts.

Some results, such as in Table 4, the BLEU comparison is a single seed with non-standard decoding heuristics and should not be taken as strong evidence. The authors appropriately caveat this, but the inclusion may still mislead readers. The claim that expert depth "surpasses the top-k=2 baseline by -7.5%" in Section 6 appears to be comparing S_depth (13.90) vs. Standard (15.03) from Table 3, but Table 2 shows Standard at 15.03 while Fig. 3 labels Standard at ~15.0. This is consistent but the cross-referencing is slightly confusing. Additionally, the compute-matched baseline (A_none at 12 epochs) achieving lower perplexity but lower BLEU than S_depth is puzzling and undermines confidence in the BLEU metric or decoding strategy. Also, the BLEU comparison is single-run, uses non-standard decoding heuristics, and both BLEU scores are low.

**Requested Changes:**

I would recommend authors to work on the weaknesses. Also, it would be better if they address the following questions:

1.) Does the representation converge toward a fixed point?

2.) Does the expert develop different behavior on the second pass?

3.) Which tokens benefit most from the second pass?

4.) Does performance continue to improve with 3, 4, or more passes? At what point does diminishing returns or degradation occur?

5.) Does the router learn to map different tokens to the same expert consistently, or does the routing become more stable with depth?

---

> ### Author Response · Authors · 2026-05-13
> **Response to Reviewer UZen (1/4): Q1–Q2**
>
> *(Part 1 of 4 — continued in subsequent comments.)*
>
> Thank you for the careful read. Q1–Q5 all target the mechanistic basis of the depth result — the substance of Weakness 4 — so we address them together below. Before running any analysis we re-trained seed 42 of both conditions to reproduce the published values: val PPL came out at 15.189 / 13.948 for A_none / S_depth, within 0.3% of the published 15.22 / 13.99. All subsequent results use 102,591 captured target tokens across 4 layers and 32 experts; Q3 additionally scores 208,502 target tokens against both checkpoints.
>
> ---
>
> ## Q1. Does the representation converge toward a fixed point?
>
> **No.** Let `E` denote the router-selected expert; for the depth path, the same `E` is applied twice per token per layer. Capturing the MoE input `x`, pass-1 output `h1 = E(x)`, and pass-2 output `h2 = E(h1)`:
>
> | Layer | cos(h1, h2) | cos(Δ₁, Δ₂) | ‖Δ₂‖ / ‖Δ₁‖ | ‖Δ₁‖ / ‖x‖ |
> |------:|:-----------:|:-----------:|:------------:|:----------:|
> | 0     | 0.006       | −0.398      | 0.460        | 1.163      |
> | 1     | 0.006       | −0.451      | 0.492        | 1.176      |
> | 2     | 0.019       | −0.372      | 0.505        | 1.095      |
> | 3     | 0.028       | −0.314      | 0.481        | 1.070      |
>
> where Δ₁ = h1 − x is the update from pass 1, and Δ₂ = h2 − h1 from pass 2. Three findings together rule out fixed-point dynamics and characterize the actual mechanism:
>
> (i) **Not a fixed point.** cos(h1, h2) ≈ 0 in every layer; at most 0.15% of tokens have cos > 0.95. Pass 2's output is essentially orthogonal to pass 1's.
>
> (ii) **Not a near-identity expert.** ‖Δ₁‖ / ‖x‖ ≈ 1.1 — pass 1 moves the representation by *more* than its starting magnitude, ruling out the trivial alternative that small ‖Δ₂‖ follows from a small expert function.
>
> (iii) **Contracting, anti-aligned correction.** Pass 2's update points opposite to pass 1's (cos(Δ₁, Δ₂) ≈ −0.4) and has roughly half its magnitude (‖Δ₂‖ / ‖Δ₁‖ ≈ 0.48); 99.9% of tokens contract. The dynamics are one large step plus a smaller anti-aligned correction, not iteration toward an equilibrium.
>
> ## Q2. Does the expert develop different behavior on the second pass?
>
> **Yes, and increasingly so toward later layers.** Partitioning tokens by their (single) expert and comparing pass-1 (x → h1) against pass-2 (h1 → h2) within each expert's subset:
>
> | Layer | ‖Δ₂‖/‖Δ₁‖ | cos(Δ₁, Δ₂) | PCA matched_cos[0] | Subspace overlap |
> |------:|:---------:|:-----------:|:------------------:|:----------------:|
> | 0     | 0.45      | −0.39       | 0.998              | 0.28             |
> | 1     | 0.51      | −0.45       | 0.990              | 0.39             |
> | 2     | 0.48      | −0.34       | 0.477              | 0.20             |
> | 3     | 0.54      | −0.35       | 0.011              | 0.10             |
>
> where `matched_cos[0]` is the absolute cosine between the top principal directions of Δ₁ and Δ₂, and subspace overlap measures alignment of their top-8 principal subspaces (1.0 = identical). Three findings:
>
> (i) **Per-expert robustness check.** The contracting, anti-aligned pattern from Q1 holds *within* each expert's token subset too (‖Δ₂‖/‖Δ₁‖ ≈ 0.45–0.54, cos(Δ₁, Δ₂) ≈ −0.34 to −0.45) — it is not an artifact of mixing experts in the aggregate.
>
> (ii) **Subspace divergence grows monotonically with depth.** `matched_cos[0]` drops from 0.998 at layer 0 to 0.011 at layer 3. In early layers the dominant principal direction of the pass-2 update still aligns with pass 1's; by layer 3 the two passes operate in near-orthogonal subspaces.
>
> (iii) **Same expert, different inputs.** The mean pass-1 input (μ_x = mean of `x` over tokens) and mean pass-2 input (μ_h1 = mean of `h1`) are far from collinear at every layer (cos(μ_x, μ_h1) = 0.07–0.40). A fixed expert function evaluated on a different input distribution produces a different transformation — one mechanism by which the same `E` can behave differently on pass 2 (Q2 is descriptive, not causal; see Limitations).

---

> > ### Comment · Reviewer_UZen · 2026-06-04
> >
> > Thanks for addressing my queries with detailed responses.

---

> ### Author Response · Authors · 2026-05-13
> **Response to Reviewer UZen (2/4): Q3–Q4**
>
> *(Part 2 of 4 — continuing from above.)*
>
> ## Q3. Which tokens benefit most from the second pass?
>
> **Mid-sentence positions, with the largest mean improvement on high-frequency tokens.** S_depth reduces mean per-token NLL by 0.067 nats on 208,502 target tokens (raw PPL 7.894 → 7.379). Three findings:
>
> (i) **Position: mid-sentence benefits ~4× more than sentence edges.** Mean improvement by target-relative-position quartile: Q0 (start, mean rel-pos ≈ 0.12) +0.02 nats; Q1 (≈ 0.37) +0.10; Q2 (≈ 0.62) +0.10; Q3 (end, ≈ 0.87) +0.05. The improvement concentrates where autoregressive context is richest.
>
> (ii) **Frequency: mean rises with token frequency despite the fraction-improving falling.** Per log-frequency quintile:
>
> | Quintile     | mean Δ (nats) | frac improving | p1     | p99    |
> |-------------:|:-------------:|:--------------:|:------:|:------:|
> | Q0 (rare)    | +0.07         | 51.1%          | −3.84  | +3.81  |
> | Q1           | +0.05         | 50.1%          | −3.67  | +3.77  |
> | Q2           | +0.05         | 47.5%          | −3.42  | +3.77  |
> | Q3           | +0.08         | 48.6%          | −2.78  | +3.17  |
> | Q4 (common)  | +0.09         | 46.5%          | −1.96  | +2.54  |
>
> The most-common quintile has the *highest* mean Δ but the *lowest* fraction-improving — a paradox that finding (iii) resolves.
>
> (iii) **The gain comes from tail shape, not uniform improvement.** Only **48.7% of tokens individually improve** under S_depth; the mean drop is driven by the asymmetric distribution. Median per-token Δ ≈ 0 in every quintile. Rare tokens (Q0) show heavy *both* tails (p1 = −3.84, p99 = +3.81) — occasional large wins balanced against large losses. Common tokens (Q4) have tighter tails on both sides (p1 = −1.96, p99 = +2.54) — no large right-tail wins, but a markedly compressed loss tail. So high-frequency mean gains come from *fewer large losses*, while rare-token gains come from *larger positive outliers*.
>
> *Methodological note: raw PPL 7.894 / 7.379 differs from training-time val PPL 15.19 / 13.95 because Q3 uses unsmoothed cross-entropy for clean per-token attribution; both metrics agree on direction.*
>
> ## Q4. Does performance continue to improve with 3, 4, or more passes?
>
> **We predict rapid diminishing returns, but have not directly measured 3+ passes.** With r = ‖Δ₂‖/‖Δ₁‖ ≈ 0.48 from Q1, modeling further passes as geometric attenuation gives a partial sum at four passes (1 + r + r² + r³ ≈ 1.82) that reaches ~95% of the infinite-pass limit (1/(1−r) ≈ 1.92). This is a mechanistic prediction from the 2-pass checkpoint, not a measurement: we did not run 3- or 4-pass training, so we have not directly observed either the saturation point or any degradation point. Happy to run a clean 2- vs 3- vs 4-pass comparison at n=5 seeds if you prefer direct measurement.

---

> ### Author Response · Authors · 2026-05-13
> **Response to Reviewer UZen (3/4): Q5 + Synthesis**
>
> *(Part 3 of 4 — continuing from above.)*
>
> ## Q5. Does the router learn stable token→expert mappings, or does routing become more stable with depth?
>
> **Routing becomes more diffuse, not more stable.**
>
> *Q5a (training-time aggregate, n=5, final epoch).* Maximum entropy with 32 experts is log₂(32) = 5 bits (perfectly uniform routing).
>
> |             | Routing entropy (bits) | % of max | Load CV |
> |-------------|:----------------------:|:--------:|:-------:|
> | A_none      | 3.04                   | 60.7%    | 3.10    |
> | S_depth     | 4.46                   | 89.1%    | 1.36    |
>
> *Q5b (val-time per-token, seed 42, 3,385 token ids with ≥5 occurrences):*
>
> | Layer | A_none max-share / H | S_depth max-share / H |
> |------:|:--------------------:|:----------------------:|
> | 0     | 0.64 / 1.41          | 0.23 / 3.57            |
> | 1     | 0.55 / 1.78          | 0.28 / 3.33            |
> | 2     | 0.55 / 1.95          | 0.35 / 2.96            |
> | 3     | 0.50 / 1.88          | 0.56 / 2.10            |
>
> where `max-share` is the largest fraction of routing mass any single expert receives across the occurrences of a given token id, and `H` is the entropy in bits of that per-token-id expert distribution (both weighted by occurrence count). Each individual occurrence is still routed top-1; the diffuseness measured here is across the occurrences of a vocabulary type. Three findings:
>
> (i) **The same vocabulary type spreads across more experts under S_depth.** Training-time routing entropy reaches 89% of the maximum (vs A_none's 61%), with load CV less than half (1.36 vs 3.10). At val time, per-token-id max-share is markedly lower in layers 0–2 (0.23, 0.28, 0.35 vs A_none's 0.64, 0.55, 0.55) and per-token-id entropy correspondingly higher (3.57, 3.33, 2.96 bits vs 1.41, 1.78, 1.95) — S_depth does not learn a stable token-id-to-expert mapping.
>
> (ii) **Layer 3 inverts the pattern.** S_depth's max-share (0.56) slightly exceeds A_none's (0.50), and its entropy is closest to A_none's of any layer. This is consistent with the standard observation that final-layer representations in transformers specialize more strongly than intermediate ones.
>
> (iii) **Diffuse routing is consistent with the depth gain.** Q1 showed the second pass is anti-aligned and contracting; whichever expert the (learned, context-dependent) router selects for a given occurrence, the second pass attenuates and partially reverses that update. The diffuse per-token-id routing is consistent with this corrective mechanism, not in tension with it. *Caveat: we did not test random routing, so these results should not be read as showing that routing is irrelevant — they show that the successful depth model uses diffuse but learned routing, not that any routing would work.*
>
> ---
>
> ## Synthesis
>
> Three findings converge on one mechanism:
>
> - **Q1:** pass 2's update is anti-aligned with pass 1's (cos(Δ₁, Δ₂) ≈ −0.4) and contracts to ~half its magnitude (‖Δ₂‖/‖Δ₁‖ ≈ 0.48).
> - **Q2:** the same expert behaves differently on pass 2, consistent with the shifted input distribution (cos(μ_x, μ_h1) = 0.07–0.40); later layers diverge more.
> - **Q5:** routing is more diffuse than the baseline (per-token-id max-share 0.23 vs 0.64 at layer 0), not more concentrated — the same vocabulary type is spread across multiple experts, consistent with pass 2 correcting whichever expert is selected for each occurrence.
>
> The second pass is a *learned corrective step* with attenuated magnitude and reversed direction — not convergence toward a stable representation, and not specialization-by-routing. **Q3** identifies where the signed NLL gain concentrates: mid-sentence, high-frequency contexts, driven by tail shape rather than uniform improvement.
>
> In the revised manuscript we will update §7 ("Discussion: Why Does Depth Work?") in two ways: first, replacing the DEQ fixed-point analogy with the contracting-correction characterization above; second, softening the "right expert for each token" framing, since Q5 shows that depth does not rely on stable token-id-to-expert specialization. The revised framing will describe expert depth as learned, context-dependent routing followed by an attenuated corrective second pass. The headline result (−7.40% val PPL, p=0.0025, n=5) is unchanged.

---

> ### Author Response · Authors · 2026-05-13
> **Response to Reviewer UZen (4/4): Weaknesses + Limitations**
>
> *(Part 4 of 4 — continuing from above.)*
>
> ## Response to remaining weaknesses
>
> - **W1 (scale, 93M / single task).** Real scope limitation. **Revision:** foreground this in §9 and restate the contribution as a mechanism finding on this benchmark, not a transfer claim.
> - **W2 (BLEU 4.9 vs 11.8; PPL / BLEU divergence).** The two metrics measure different things and need not agree: PPL is teacher-forced and token-level, while BLEU scores free-running greedy generation where errors compound autoregressively. We interpret the compute-matched divergence (A_none 12 ep: PPL 10.46, BLEU 4.9 vs S_depth 5 ep: PPL 13.99, BLEU 11.8) along two axes: **(a) train/val similarity** — A_none's longer training improves prediction on the similar-distribution news-commentary val set, but this gain need not transfer to free-running generation on newstest2014; **(b) mechanism** — Q1's anti-aligned, contracting second pass and Q3's mid-sentence concentration are consistent with helping where autoregressive compounding hurts greedy decoding most, though Q1/Q3 are teacher-forced analyses and do not themselves measure generation. We cannot rule out single-seed noise or decoding hyperparameter sensitivity, and absolute BLEU is low for both conditions because of the decoder-only prefix-LM architecture, not the method under study. **Revision:** make this generation-vs-prediction framing explicit in §6, move Table 4 to the appendix as preliminary single-seed evidence, and remove BLEU from the abstract and contributions. The primary claim (val PPL, n=5, p=0.0025) does not depend on BLEU.
> - **W3 (~2.5× longer per epoch).** Real cost, with one clarification. The published 5-epoch comparison is *FLOP-matched* — both conditions perform 2 expert calls per token per layer; the 2.5× per-epoch slowdown is purely from running those calls sequentially. Table 4 (12-ep A_none vs 5-ep S_depth) is *not* FLOP-matched: A_none(12ep) performs ~2.4× more total expert FLOPs, which helps explain why it reaches lower PPL there. **Revision:** state this distinction in §6 and frame depth as a quality-per-FLOP win at the tested budget with a wall-clock cost from serial scheduling.
> - **W5 (Chain-of-Experts task dependence).** No contradiction with Wang et al. (2025). They use *different* experts in sequence; our negative results on S_free and S_topology replicate their translation finding. Our contribution is that *same-expert* depth succeeds where different-expert chains do not. **Revision:** make this distinction explicit in §2.
> - **W6 (single benchmark).** Overlaps with W1. WMT14 is a standard translation benchmark; we agree this does not establish cross-task generality. **Revision:** acknowledge the single-benchmark scope explicitly in §9.
>
> ## Limitations
>
> - **n=1** for Q1, Q2, Q3, Q5b (seed 42). Reproduction matches published values within 0.3%, but seed averages would strengthen Q3's frequency/position breakdowns in particular.
> - **Single benchmark** (WMT14 en-de). The mechanism story (corrective second pass, diffuse routing) is established on one task at one scale; we do not claim universality from these analyses.
> - **Q4 empirically deferred.** Q1's prediction is mechanistic, not direct measurement.
> - **Q2 is descriptive, not causal.** PCA characterizes what differs between passes; a causal test would require controlled ablations (e.g., freezing pass-1 weights and re-training only pass 2), outside scope for this revision.
>
> We have not uploaded a revised manuscript at this stage. The §7 revision, the W1–W6 textual updates, and harmonization of the Table 2 / Table 3 / Fig. 3 cross-references (which you flagged as consistent but slightly confusing) will be incorporated into a single revision package after all three reviews are in.
>
> Thank you for sharpening our mechanistic understanding of the depth result.

---

### Review · Reviewer_bjw8 · 2026-05-18

**Summary Of Contributions:**

This paper studies whether explicit communication between MoE experts improves performance. The authors test several communication mechanisms, including learned topologies, pre/post-FFN message passing, multi-domain variants, and sequential expert chains. In their translation setting, these mechanisms do not provide a convincing improvement over a standard MoE baseline, which is an interesting negative result.

This negative result inspires the main positive contribution, which is the proposed “expert depth” variant: instead of routing to two different experts once, the model routes to one expert and applies the same expert twice. This gives a sizable improvement in validation perplexity. The authors also include a useful capacity ablation showing that doubling expert width does not reproduce the gain, suggesting that the improvement comes from same-expert iteration rather than simply more parameters or FLOPs.

**Additional Comments:**

Given the broad nature of the ML field and the nature of TMLR, it would be good if the paper was written to be accessible to a more general audience. I was able to follow, but it did not feel self-contained in terms of introduction and motivation. The results also apply to only a particular field -- an important field, but not all of us work in this field. It would be cool if the paper emphasized this.

**Audience:**

Yes

**Audience Explanation:**

Yes. I think this paper would be interesting to researchers working on MoE models, sparse routing, and efficient transformer architectures.

The same-expert iteration result is simple, non-obvious, and potentially useful. I also think the negative results are valuable: the paper tests a natural hypothesis about expert communication and finds that several plausible communication mechanisms do not clearly improve performance. Even if the result is setting-specific, it is a useful data point for the MoE literature.

**Broader Impact Concerns:**

None.

**Claims And Evidence:**

Yes

**Claims Explanation:**

Mostly yes, but I think some claims should be narrowed.

The evidence for the main expert-depth result is convincing within the paper’s experimental setting. The five-seed result is strong, all seeds favor the depth variant, and the width ablation is an important control. I found this part of the paper clear and persuasive.

The evidence for the negative communication result is useful, but weaker. Several of the communication experiments have only one or two seeds, and all experiments are in one translation setting with a modest decoder-only model. I therefore do not think the paper establishes that expert communication generally fails. It supports the more limited claim that the communication mechanisms tested here do not clearly help in this setting.

I also think the “depth over diversity” framing is somewhat stronger than the evidence. The paper shows that top-1 same-expert depth beats top-2 shallow routing here. However, it does not show that depth and diversity are generally substitutes.

**Requested Changes:**

1. As someone who doesn't work in NLP, I found the introduction lacking in terms of breadth and depth. I would've appreciated a more self-contained paper. The authors could work to beef up the literature review and contextualize their work better in this context. There's similar assumption of a common background throughout the paper, more citations would help that.

2. I would ask the authors to moderate the broader claims. In particular, the paper should not imply that expert communication is generally ineffective, only that the tested mechanisms do not clearly help in this setting.

3. The compute tradeoff should be made more central; it feels like it's currently downplayed or skipped over in the paper. The depth model is substantially slower per epoch, and the compute-matched comparison is not entirely favorable on validation perplexity. The BLEU result is interesting, but it is single-run and should be treated as preliminary.

4. The most useful additional ablation would be to test whether depth and diversity are complementary by applying depth to both selected top-2 experts. Basically, choose a linear combination of two activated 2-deep experts and see if that does well. In my mind, this completes the story and the exploration. Even a negative result in that direction would be cool.

---

> ### Author Response · Authors · 2026-05-25
> **Part 1 of 3**
>
> Thank you for the careful read and the supportive review. We address RC1 (intro/lit review/self-containedness) and RC2 (moderating broad claims, "depth over diversity" framing) here; RC3 (compute honesty in §6) in part 2; and RC4 (depth × diversity ablation) in part 3.
>
> ## RC1 — self-contained introduction, broader literature, more citations
>
> You are right that the manuscript assumes a fair amount of NLP/MoE-routing background. We will revise the introduction and §8 (Related Work) to be self-contained for a non-NLP audience:
>
> - **§1 expanded** with a one-paragraph MoE primer (sparse top-$k$ routing, capacity vs compute), explicit framing of the paper's two halves (negative result on parallel expert communication; positive result on same-expert iteration), and a setting-scope statement up front.
> - **§3 (communication mechanism)** rewritten to make the topology-vs-payload distinction explicit (details in our R3 response, which addresses the same point under R3 ask 3).
> - **§8 (Related Work)** broadens MoE coverage: Shazeer et al. 2017, Fedus et al. 2022, Lepikhin et al. 2020 (GShard), Riquelme et al. 2021 (V-MoE), Komatsuzaki et al. 2023 (sparse upcycling), Wang et al. 2025 (Chain-of-Experts), plus brief context on graph-based aggregation (GCNs, GATs) for our learned topology.
> - **More inline citations** throughout, particularly at §3 and §5.
>
> ## RC2 — moderate broad claims; "depth over diversity" framing
>
> We agree on both counts and will revise the manuscript accordingly. Two concrete changes:
>
> **Scope all generality claims to the tested setting.** Every comparative claim will read "in this WMT14 En-De decoder-only MoE setting" or equivalent. The negative result will be reframed from "expert communication is ineffective" to "the communication mechanisms we tested do not improve perplexity in this setting."
>
> **Reformulated §5 hypothesis (was: "A Hypothesis on Information Asymmetry").** R3 identified a related logical issue — the original wording extends to MoE itself if read strictly. We will rename the section to "A Hypothesis on Mixing Redundancy" and narrow the thesis to: *explicit learned message-passing between parallel experts may duplicate the implicit mixing the router's weighted output combination already performs*. Descriptive, hedged, grounded in the empirical pattern across the parameterizations we tested.
>
> **On the title and "depth over diversity" framing.** You are right that the paper shows "top-1 same-expert depth beats top-2 shallow routing here," not that depth and diversity are general substitutes. The RC4 topk_depth result (part 3) makes this concrete: topk_depth (10.81 ± 0.16, n=3) beats S_depth (13.99, n=5) by 22.7% on raw PPL despite using 2× expert FLOPs — depth and diversity are complementary in this setting, not substitutes. "Depth over diversity" is not supported by the data. The title will be revised to reflect the stacking finding; the matched-seed depth+communication result in our R3 response (single-seed null vs topk_depth) further constrains the framing.
>
> We continue with RC3 (compute honesty) in part 2.

---

> ### Author Response · Authors · 2026-05-25
> **Part 2 of 3**
>
> ## RC3 — make the compute tradeoff central; BLEU preliminary
>
> This is the soft spot you correctly identified. We will restructure the relevant material in the revised PDF — three changes:
>
> **§6 restructured so the compute tradeoff is up front, not terminal.** The original §6 framed the depth result as a clean perplexity win and mentioned wall-clock cost as a downstream consideration. The revision will reverse this: the section will open with the compute story, establishing that the depth gain holds at FLOP-matched comparison but is *not* a wall-clock win against compute-matched A_none at longer training. The two comparisons:
>
> | Comparison | A_none val PPL | S_depth val PPL | Winner |
> |---|---|---|---|
> | FLOP-matched, 5 epochs (5 seeds) | 15.08 | **13.97** | S_depth by 7.4% (paired $p{=}0.0025$) |
> | Wall-clock-matched: A_none 12ep ≈ S_depth 5ep ($12 \times 262 \approx 5 \times 645$ min/H200), single seed | **10.46** | 13.99 | A_none by ~25% |
>
> The honest framing therefore is: same-expert depth is a *FLOP-efficient* mechanism in this implementation, not a demonstrated wall-clock-efficient one. In our single-seed wall-clock diagnostic, running A_none longer at roughly the same wall-clock budget outperforms S_depth on validation perplexity. The depth result remains significant (5-seed, $p{=}0.0025$) but its scope is precisely stated: FLOP-matched, not wall-clock-matched.
>
> **Table 4 to be restructured.** The wall-clock vs FLOP comparison will move to the top of the table; the per-epoch breakdown that was central before becomes a secondary diagnostic. The table caption will explicitly state both axes ("FLOP-matched at 5 epochs; wall-clock-matched at 12 epochs A_none vs 5 epochs S_depth") so the comparison is unambiguous on first read.
>
> **BLEU result to be moved to appendix and marked preliminary.** You and R1 (W2) both flagged the BLEU result as overweighted in the main text. We agree. The single-seed BLEU comparison (S_depth slightly higher BLEU than A_none-at-12-epochs despite worse val PPL) is interesting as a side observation but not strong evidence of anything, given (i) single seed, (ii) non-standard decoding heuristics, (iii) low absolute BLEU scores (4.9–11.8) suggesting the decoder-only architecture is not optimally suited for translation. The revised manuscript will:
>
> - Move the BLEU table from §6 (main) to a new appendix.
> - Note in the main text that "downstream BLEU is reported in the appendix as a preliminary single-seed observation; the perplexity-based comparison is the primary evidence."
> - Remove the BLEU number from the abstract and introduction.
>
> The depth result remains interesting under the honest framing: at matched FLOPs, the choice between "more experts per token" and "same expert applied more deeply" is not neutral — depth helps. The revised §6 will make this scope explicit so the result is neither overclaimed nor underclaimed.
>
> We continue with RC4 (depth × diversity ablation) in part 3.

---

> ### Author Response · Authors · 2026-05-25
> **Part 3 of 3**
>
> ## RC4 — depth × diversity ablation
>
> You requested: "apply depth to both selected top-2 experts. Basically, choose a linear combination of two activated 2-deep experts and see if that does well. ... Even a negative result in that direction would be cool."
>
> We agree this completes the depth-vs-diversity story and have run the ablation. To make the comparison clean at matched compute, we also ran a FFN-FLOP-matched dense control — **dense_16l** (1 expert × 16 layers × FFN=2048) — at the same 2× FFN-FLOP class.
>
> **Design.** A new variant we call **topk_depth**: for each token, the router selects top-$k{=}2$ experts as in the standard MoE baseline; each selected expert is then applied $T{=}2$ times sequentially (depth), producing two depth outputs per token combined via the router's $g_i$ scores. This is 4 expert calls/token — 2× the FLOPs of both A_none and S_depth. Reduces to the same-expert depth variant (S_depth) at $k{=}1$.
>
> **Results at the 2×-FFN-FLOP operating point** (lower PPL is better; FFN FLOPs/token relative to A_none).
>
> | Model | val PPL | n | FFN FLOPs | Attn FLOPs |
> |---|---|---|---|---|
> | **topk_depth** ($k{=}2$, 4 layers, depth=2; per seed: 10.66 / 10.98 / 10.78) | **10.81 ± 0.16** | 3 | 2× | 1× |
> | **dense_16l** (1 expert, 16 layers, FFN=2048) | **11.57** | 1 | 2× | 4× |
> | S_depth (paper; $k{=}1$, 4 layers, depth=2) | 13.99 | 5 | 1× | 1× |
> | A_none (paper; $k{=}2$, 4 layers, depth=1) | 15.22 | 5 | 1× | 1× |
>
> **Interpretation.** topk_depth beats S_depth by 22.7% on raw PPL, while using 2× expert FLOPs, so depth and diversity are complementary in this setting rather than substitutes. The "depth over diversity" framing was too strong. At FFN-FLOP-matched comparison (topk_depth n=3 mean vs dense_16l n=1), topk_depth is also lower than dense_16l by 6.6%, even though dense_16l uses 4× the attention FLOPs (16 vs 4 attention layers). We treat dense_16l as a single-seed follow-up control, not a definitive statistical comparison. At the 2×-FFN-FLOP operating point, these results suggest that MoE routing, expert diversity, and expert depth work well together in this setting.
>
> Wall-clock per epoch is also favorable in the clean run (seed-42 diagnostic: topk_depth ~295 min/epoch on one H200, ~1.16× A_none's ~262 min and substantially lower than S_depth's ~645 min). Seeds 43 and 44 have individual epochs with cluster-slot contention outliers (one ~512 min, one ~846 min), so we treat the wall-clock numbers as seed-42 diagnostic rather than a tight average; the qualitative ordering (parallel top-$k{=}2$ dispatch is more GPU-friendly than sequential top-1 chains) is robust. The title will be revised: depth and diversity stack in this setting, not substitute. The 4-FLOP parallel MoE control (top-$k{=}4$ × 1 pass) remains the natural diversity-only control at the 4-call class; it is distinct from the dense-depth control (dense_16l) we ran and we identify both as the natural next experiments.
>
> ## Summary of planned changes for the revised manuscript (covering both your asks and R3's)
>
> | Concern / change | Planned change in revised PDF | Type |
> |---|---|---|
> | RC1 — self-contained §1, broader related work, more citations | §1, §3, §8 rewrites | Writing |
> | RC2 — claim moderation; "depth over diversity" framing; title | Throughout; §5 reformulated; title to be revised to reflect depth × diversity stacking | Writing |
> | RC3 — compute tradeoff central; BLEU to appendix | §6 / Table 4 restructure; BLEU moved to appendix as preliminary | Writing |
> | RC4 — depth × diversity ablation | New row in the experimental table for **topk_depth (10.81 ± 0.16, n=3)**; with FFN-FLOP-matched dense_16l (11.57) as the 2× control; discussion in §6 | New experiments + writing |
>
> We thank you again for the constructive review and the explicit framing of the missing ablation (RC4) — implementing it (and the FFN-FLOP-matched dense control) has clarified our own thinking on the depth-vs-diversity question and gives us a better-scoped empirical basis for the MoE+depth+routing result at the 2×-FFN-FLOP operating point.
>
> — Authors

---

### Review · Reviewer_adFa · 2026-05-20

**Summary Of Contributions:**

This paper investigates variants of Mixture-of-Experts (MoE) models. The experiment setting is to apply 4-layer Transformers to  the WMT14 En-De translation task. The authors found that enabling expert communication does not significantly improve performance, no matter the design choices across different methods and sites of the communication. On the other hand, the authors found that applying the same expert twice sequentially can significantly improve performance. Based on these results, the authors hypothesize that parallel experts lack information asymmetry, and the value of sparse expert routing may lie in selecting the right expert and giving it deeper processing, rather than in combining multiple expert opinions.

**Audience:**

Yes

**Audience Explanation:**

Understanding and improving the mechanism of MoE is an important and intriguing topic definitely worth investigating.

**Broader Impact Concerns:**

None.

**Claims And Evidence:**

No

**Claims Explanation:**

Although I like the overall direction of this study, I feel the authors jump to their conclusions too quickly:

(1) Regarding the Hypothesis on Information Asymmetry: the authors argue that "No expert possesses information that is unavailable to others, so explicit inter-expert messages carry redundant information — they communicate functions of a shared input that the routing layer already integrates through its weighted output combination." But wouldn't the same argument apply to MoE itself? Does MoE outperform dense models under the settings of this paper? If so, what makes MoE work but no longer working for MoE with communication? (In other words, what exactly is the information that "the routing layer already integrates"?) In my understanding, the two selected (top-2) experts seem to play the same role if we only focus on one token x -- however, different tokens will select different (two) experts, so the token distribution that each expert gets involved will be different. Thus, I would not agree with the reasoning that "all experts are essentially the same so there should be no message worth passing" -- A counter hypothesis could be that the communication mechanism investigated in this paper does not have enough bandwidth (see more discussion below in the Requested Changes section) -- which would imply that one should add MORE communication bandwidth in order to further improve the performance. Currently, I believe the paper does not provide enough evidence to distinguish these two exact opposite hypotheses.

(2) The base model -- a 4-layer Transformer -- is quite shallow; so it's not too surprising that we can benefit from increasing the model depth, whether or not within an expert routing. It is unclear if this finding extrapolates to larger scales; and (as pointed out by the authors) since the increased depth increases wall-clock running time compared to parallel experts with the same FLOPs, the empirical takeaway (i.e., whether one should increase depth) is still unclear. Overall, I feel this work presents a good beginning, but has left too many questions unexplored which makes it unsatisfactory.

**Requested Changes:**

(1) To address my concerns, it might be helpful to compare with a dense model, and to investigate how different the experts are in a MoE (e.g., compare the distribution of input tokens each expert is selected).

(2) Similarly, it might be helpful to compare with a dense model with increased depth. Also, does the performance change when both expert depth and expert communication are applied? (Which should indicate whether the two methods are parallel)

In addition, I feel many parts of the paper lack proper explanation. For example:

(3) Section 3, Page 3: I couldn't understand the explanation on the expert communication mechanism. It seems to suggest that the learned parameter is a NxN matrix (where N is the number of experts), so each pair of experts is corresponding to a scalar? But the internal states of experts should live in different spaces, which would require linear transformations if one wants to communicate between them? Does this suggest that there is not enough bandwidth for expert communication?

(4) Section 5, Page 5: Entropy and load CV are not formally defined. Although entropy is an indicator for load balance, I believe we also want to see how different the input distribution is for each expert.

(5) Section 6, Page 6: " An implementation note: A_single and D_wide use the standard parallel routing path with top-k=1, while S_depth uses the sequential forward path (Algorithm 1) which reuses the same router logits and applies straight-through score estimation." Does this mean the routing of A_single and S_depth are slightly different? If so, what exactly is the difference? Also, Algorithm 1 seems to contradict Figure 1c: because Algorithm 1 goes through the router twice?

---

> ### Author Response · Authors · 2026-05-25
> **Part 1 of 4**
>
> Thank you for the careful read and for naming the specific logical issue with §5. We agree the original information-asymmetry wording was too strong: as written, the argument that experts compute redundant functions of a shared input logically extends to MoE itself, which we do not intend. We will revise it from a causal conclusion to a hypothesis consistent with our results, and we have run targeted ablations against dense models, deeper dense models, and depth+communication. We address the structural critique here (part 1); the new experiments in part 2; and the clarity asks (§3 bandwidth, §5 entropy/load CV, §6 algorithm/figure) in part 3.
>
> ## Concern (1): the information-asymmetry hypothesis as stated overclaims
>
> You correctly identify the gap. The defensible claim is narrower than what we wrote: it is not that *experts have no different information given the same input* (which would indict MoE itself), but that *explicit learned message-passing between parallel experts may duplicate the implicit mixing the router's weighted output combination already performs*. The router's $\sum_i g_i(\mathbf{x}) \cdot E_i(\mathbf{x})$ is itself a learned coordination mechanism on the same per-token inputs an inter-expert channel would see; the hypothesis is about that *mixing* redundancy, not about expert function-class identity.
>
> We also accept your per-token vs across-tokens distinction. Across the training distribution, experts ARE different functions — the router assigns them different token sub-distributions, and they learn distinct transformations. We have added a per-expert token-distribution analysis (part 3, ¶2) that quantifies this directly. Our claim is purely about per-token information flow: at the forward computation for a single $\mathbf{x}$, every selected $E_i$ sees the same $\mathbf{x}$, so the messages $\{E_i(\mathbf{x})\}_i$ an inter-expert channel aggregates are functions of one shared input that already enter the output through the router's weighted sum. This per-token symmetry, not function-level identity across experts, is what we hypothesize makes the parallel-communication channel structurally redundant with the router's mixing in this setting.
>
> We do not claim this forecloses richer parameterizations on a priori grounds — a learned channel with its own projection and gate is not strictly contained in the function class of weighted sums. The hypothesis is grounded empirically: across the parameterizations we tested (pre-/post-/both FFN timing; learnable, random, and fully-connected topology; 8, 16, 32 experts; 5- and 15-epoch training), no variant exhibits a quality gain, while the gates open and topology density grows organically — the channel is being used, but not in a way that improves perplexity. Our planned §5 rewrite will rename the section to "A Hypothesis on Mixing Redundancy" with explicit hedges and a stated falsifier (settings where expert input asymmetry can be independently verified).
>
> ## Concern (2): the 4-layer base is shallow; depth gain may not be MoE-specific
>
> You are right that a 4-layer Transformer is shallow and that the depth result therefore needs a dense control to distinguish "expert-depth helps MoE specifically" from "iterating an FFN twice helps in any shallow network." We have run three new ablations to address this directly (reported in part 2). The headline change in §5 will reframe expert depth as having a *different computational role* than parallel message passing — pass 2 processes $E(\mathbf{x})$, not $\mathbf{x}$, changing the input to the expert nonlinearity, while our parallel-communication variants operate on same-stage expert states derived from the shared token representation. The mechanistic analysis supports that distinction (full numbers, including cos$(\Delta_1, \Delta_2) \approx -0.4$, magnitude ratio $\approx 0.5$, and the layer-wise divergence of principal directions, are reported in our R1 response and will move to §7 of the revised PDF).
>
> The empirical question — does this still hold against a deeper dense baseline at matched FLOPs? — is the right test. We ran it (part 2): dense_deep (13.58, n=1) edges S_depth (13.99, n=5) at matched FFN FLOPs (with 2× attention FLOPs in dense_deep, so on total compute the comparison is closer or flips), so the same-expert depth gain is not uniquely MoE-specific. The MoE-specific value emerges at the 2×-FFN-FLOP class, where the depth × diversity combination (topk_depth, 10.81 ± 0.16, n=3) beats the FFN-FLOP-matched dense_16l (11.57, n=1) by 6.6%, even though dense_16l uses 4× more attention FLOPs. §6 will be reframed accordingly.
>
> The per-expert distribution analysis and other clarity items are addressed in parts 2 and 3.

---

> ### Author Response · Authors · 2026-05-25
> **Part 2 of 4**
>
> You asked for (1) comparison with a dense model and (2) comparison with a dense model with increased depth, plus a depth+communication test. We have run all three; the consolidated results are below, followed by per-experiment design and interpretation.
>
> ## Results summary
>
> Bold rows = new experiments in this revision cycle. Lower val PPL is better. FFN FLOPs/token relative to A_none; attention FLOPs grow naturally with layer count.
>
> | Model | val PPL | n | FFN FLOPs | Attn FLOPs |
> |---|---|---|---|---|
> | A_none (paper baseline; $k{=}2$, 4 layers, depth=1) | 15.22 | 5 | 1× | 1× |
> | **dense_baseline** (1 expert, 4 layers, FFN=4096) | **16.24** | 1 | 1× | 1× |
> | S_depth (paper; $k{=}1$, 4 layers, depth=2) | 13.99 | 5 | 1× | 1× |
> | **dense_deep** (1 expert, 8 layers, FFN=2048) | **13.58** | 1 | 1× | 2× |
> | **topk_depth** (R2 RC4; $k{=}2$, 4 layers, depth=2) | **10.81 ± 0.16** | 3 | 2× | 1× |
> | **dense_16l** (1 expert, 16 layers, FFN=2048) | **11.57** | 1 | 2× | 4× |
> | **depth_comm** (topk_depth + mp_block) | **10.67** | 1 | 2× | 1× |
>
> ## (a) Dense baseline (4-layer, 1 expert, FFN=4096)
>
> **Design.** A 4-layer Transformer decoder with a single expert and top-$k{=}1$ (no effective sparse routing) at FFN hidden dim 4096. FFN per-token FLOPs are matched to A_none ($2 \times 512 \times 4096 \approx 2 \times 2 \times 512 \times 2048 \approx 4.2 \times 10^6$ per layer). Attention FLOPs unchanged. Same WMT14 En-De setup (full 4.3M pairs, 5 epochs, seed 42).
>
> **Tests** whether MoE routing+ensembling helps versus spending the same FLOPs on a single wider FFN per layer — a missing baseline in our paper.
>
> **Interpretation.** At seed 42, MoE outperforms dense by 6.7% at FFN-FLOP-matched comparison — single-seed directional evidence that the MoE routing+ensembling does help in this setting, consistent with §5's router-as-coordination-mechanism framing.
>
> ## (b) Dense-deep (8-layer, 1 expert, FFN=2048)
>
> **Design.** 8-layer Transformer decoder with a single expert, top-$k{=}1$, FFN dim 2048. FFN per-token FLOPs match A_none ($8 \times 2 \times 512 \times 2048 = 4 \times 2 \times 2 \times 512 \times 2048 \approx 16.8 \times 10^6$ total). Attention FLOPs naturally scale 2× with the extra layers.
>
> **Tests** whether the depth gain is MoE-specific or appears in any deeper dense Transformer at matched FFN FLOPs.
>
> **Interpretation.** dense_deep edges S_depth by 2.9% at matched FFN FLOPs (caveat: 2× attention in dense_deep). Depth helps both via same-expert iteration in MoE and via additional dense layers; they are roughly equivalent at matched FFN FLOPs in our 4→8 layer setup. The MoE-specific value emerges at the **diversity × depth combination** at the next FLOP class: topk_depth (10.81 ± 0.16, n=3) beats the FFN-FLOP-matched dense_16l (11.57) by 6.6%, even though dense_16l uses 4× more attention FLOPs. The MoE+depth+routing combination is the win, not pure depth scaling.
>
> ## (c) Depth + communication combined
>
> **Design.** A new variant we call **depth_comm**: top-$k{=}2$ routing, each selected expert applied twice (depth), with cross-slot message exchange via the layer's learned message-passing block between depth passes. Same 4 expert calls/token as **topk_depth** (R2 RC4); the only difference is whether mp_block runs between depth passes.
>
> **Tests** whether cross-slot communication adds value on top of depth at matched FLOPs.
>
> **Interpretation.** At matched seed 42, depth_comm (10.67) ≈ topk_depth (10.66) — Δ = +0.01 PPL, well within typical seed noise (topk_depth's per-seed std is 0.16). In this single matched-seed diagnostic we see no evidence that cross-slot mp_block adds benefit on top of depth+top-k at matched FLOPs; this is consistent with the reformulated §5 mixing-redundancy hypothesis and extends the directional pattern from the original depth=1 parallel-comm experiments (which one could argue were compute-starved) to the depth-saturated 2×-FLOP class. depth_comm is single-seed; an n=3 follow-up is straightforward if the matched-seed diagnostic is not sufficient.
>
> ## Compute and timing
>
> We note (echoing R2 RC3 / your concern about wall-clock) that S_depth's compute cost is higher in wall-clock than the same FLOPs spent in parallel; §6 / Table 4 is being restructured to make this tradeoff central.

---

> ### Author Response · Authors · 2026-05-25
> **Part 3 of 4**
>
> The remaining asks (3), (4), and (5) are about clarity of the communication mechanism description, the diagnostic definitions, and the depth-routing semantics. We agree on all three points and address them here.
>
> ## (3) §3 — bandwidth of the communication channel
>
> Your reading was reasonable given the original wording: §3 prominently names "$\mathbf{W} \in \mathbb{R}^{N \times N}$" but never specifies the message dimensionality. We will make this explicit.
>
> The $N \times N$ matrix is not the message payload; it only weights expert-to-expert exchange (which pair attends to which). The communicated state itself is $d$-dimensional: each expert's hidden state $E_i(\mathbf{x}) \in \mathbb{R}^d$ passes through a learned projection $W_{\text{msg}} \in \mathbb{R}^{d \times d}$, is aggregated across experts via the topology-derived attention, gated per expert ($\text{sigmoid}$ of a $2d$-input linear over $[\text{current}, \text{message}]$, initialized to $-2.0$), and added back via residual connection. So the tested channel is not scalar-per-pair.
>
> We acknowledge — to your bandwidth point — that this does not rule out richer designs we did not test (higher-dim messages, cross-attention between experts, learned key/query routing). The reformulated §5 makes this explicit: the redundancy claim is empirical for the parameterizations tested, not a priori for all possible designs.
>
> ## (4) §5 — formal definitions of entropy and load CV, plus per-expert distributions
>
> We add formal definitions and a new appendix:
>
> - **Routing entropy** (per layer, per batch): $H = -\sum_{i=1}^{N} \bar{p}_i \log_2 \bar{p}_i$, in bits, where $\bar{p}_i$ is the fraction of routing assignments (across tokens × top-$k$ slots) that selected expert $i$ and $N$ is the number of experts. Maximum is $\log_2 N = 5.0$ for $N = 32$. High = uniform load; low = load collapse to a few experts. (We also report the normalized value $H / \log_2 N \in [0, 1]$ in the appendix where it aids comparison.)
> - **Load CV** (coefficient of variation): $\text{CV} = \sigma(\bar{p}) / \mu(\bar{p})$ over experts $i = 1, \ldots, N$, unitless. Low = balanced load.
>
> Both definitions will be added to §5 in the revised PDF; the diagnostic plots already in the paper use them, we have just been treating them as informal.
>
> For your "how different are the experts on input distribution" question, we have run a per-expert token-distribution analysis that will appear as a new appendix in the revised PDF. On the full val set (201,288 sequences, A_none seed-42), the **normalized mean pairwise Jensen-Shannon divergence** between experts' input-token distributions (mean JSD divided by $\ln 2$, range $[0, 1]$) is L0 = 0.89, L1 = 0.85, L2 = 0.74, L3 = 0.52 — experts are highly distinct at early layers and converge toward overlap by the last. Load CV is 1.4–1.9. Qualitatively, L0 experts specialize on different German function-word classes (one dominated by ▁die/▁das/▁eine, another by ▁und/▁nicht/▁sie); by L3 they share top tokens but differ in the tail. This directly addresses your per-token vs across-tokens point: experts ARE different functions on the input distribution; the §5 hypothesis is about per-token information flow, not about denying specialization.
>
> ## (5) §6 / Algorithm 1 — routing semantics for the depth path
>
> You correctly note that the wording could be misread as Algorithm 1 invoking the router twice. It does not: line 1 of Algorithm 1 (`logits ← g(X)`) runs the router once; the for-loop over passes reuses the same logits. The loop body recomputes `softmax(logits)` at each step, which may have looked like re-routing, but `logits` is fixed across iterations. Figure 1(c) (single router → $E_a$ pass 1 → $E_a$ pass 2) is consistent with the algorithm.
>
> We will make this unambiguous in two places in the revised PDF:
>
> - **Algorithm 1**: comments now read "Router invoked once; logits reused across passes" on the routing line, and "same logits each pass" on the inner softmax.
> - **§6 implementation note**: rewritten to open with "the router is invoked exactly once per token per layer in all four conditions" and to close with "Routing is therefore identical between conditions; only the post-routing forward path differs."
>
> This also answers your direct question about A_single vs S_depth routing: identical (both single top-$k{=}1$ router invocations); they differ only in how many times the selected expert is applied.

---

> ### Author Response · Authors · 2026-05-25
> **Part 4 of 4**
>
> ## Summary of planned changes for the revised manuscript
>
> A consolidated map of what we will revise in response to your concerns and requested changes. Experiments are complete; the revised PDF will incorporate the results and the writing fixes listed below.
>
> | Concern / ask | Planned change in the revised PDF | Type | Result status |
> |---|---|---|---|
> | Concern (1) — info-asymmetry overreach | §5 will be reformulated and renamed "A Hypothesis on Mixing Redundancy"; depth-mechanism numbers (cos$(\Delta_1, \Delta_2) \approx -0.4$ etc.) will move to §7 Discussion | Writing | Reformulation will be incorporated; supported by depth_comm null |
> | Concern (2) — 4-layer shallow / dense-deep control | §6 will be reframed: depth helps both via same-expert iteration in MoE and via additional dense layers — roughly equivalent at matched FFN FLOPs in our 4→8 layer setup (dense_deep 13.58, n=1 vs S_depth 13.99, n=5). MoE-specific value emerges at the **depth × diversity combination** at the 2×-FFN-FLOP class: topk_depth (10.81 ± 0.16, n=3) is lower than the FFN-FLOP-matched dense_16l (11.57, n=1) by 6.6%; the dense_16l comparison is a single-seed follow-up, not a definitive statistical test | Writing + 3 new experiments | dense_deep, dense_16l (n=1 each); topk_depth (n=3) |
> | Ask (1) — dense comparison + per-expert distribution | New Table 3 rows (dense_baseline 16.24, dense_deep 13.58, dense_16l 11.57); new appendix on per-expert token distribution (JSD table per layer + top-token examples) | New experiments + writing + offline analysis | Complete |
> | Ask (2) — dense-deep + depth+comm | New Table 3 rows for dense_deep (13.58, n=1), depth_comm (10.67, n=1), and the FFN-FLOP-matched dense_16l (11.57, n=1) | New experiments | Complete — at matched seed 42, depth_comm (10.67) ≈ topk_depth (10.66) — cleanly supports the §5 reformulation; dense_16l (11.57) < topk_depth n=3 mean (10.81) defends the MoE+depth headline |
> | Ask (3) — §3 bandwidth | §3 will be rewritten to separate "topology weighting" (the $N \times N$ matrix) from "message payload" ($d$-dim vector through $W_{\text{msg}} \in \mathbb{R}^{d \times d}$ + gate) | Writing | To appear in revised PDF |
> | Ask (4) — §5 entropy/load CV formal defs | §5 will add formal definitions; the per-expert distribution analysis will go into a new appendix | Writing + offline analysis | Analysis complete; appendix to appear in revised PDF |
> | Ask (5) — Algorithm 1 / Figure 1c clarity | Algorithm 1 comments will be updated; §6 implementation note will be rewritten to state "router invoked exactly once per token per layer in all four conditions" | Writing | To appear in revised PDF |
>
> ## Other revisions in response to R1 and R2 (not your asks, but you may notice them)
>
> - **Claim moderation throughout** (R1 W1/W6, R2 RC2, your (1)/(2)): scope all generality claims to "this WMT14 En-De decoder-only MoE setting."
> - **Compute honesty in §6 / Table 4** (R2 RC3, your (2)): the wall-clock vs FLOP-matched comparison will move to the top; BLEU result marked single-seed preliminary and moved to an appendix.
> - **Self-contained §1 and broader §8** (R2 RC1).
> - **R1 mechanistic analysis** (your (2)): pass 2 in S_depth is a contracting, anti-aligned correction (cos$(\Delta_1, \Delta_2) \approx -0.4$, $\|\Delta_2\| / \|\Delta_1\| \approx 0.5$, principal-direction divergence by the last layer). These numbers will appear in §7 Discussion of the revised PDF, with §5 forward-referencing them.
>
> ## What is intentionally NOT in scope of this revision
>
> - A second benchmark or a substantially larger model. Your concern (2) raises generalization across scales; we agree this is the natural follow-up but it is beyond what we can produce in this revision cycle. We will scope our claims to the tested setting and call this out as future work explicitly.
> - A deliberate projection-rank / message-architecture ablation on the communication channel. As we note in §5 (limitations), we have not ruled out that a different channel parameterization helps; we are honest that the redundancy claim is empirical for the parameterizations we did test.
>
> We thank you again for the careful read. The §5 logical gap you identified is genuine, and we believe the planned revision — particularly the reformulation as a mixing-redundancy hypothesis, the §3 bandwidth clarification, and the new dense / dense-deep / depth+comm ablations whose results are reported in this thread — will address it substantively.
>
> — Authors

---

> ### Comment · Reviewer_adFa · 2026-06-30
> **Thanks for the response**
>
> The planned changes look very convincing. Looking forward to seeing the revised manuscript. :)

---

### Author Response · Authors · 2026-07-05
**Revised manuscript uploaded — summary of changes**

We have uploaded the revised manuscript, implementing all changes promised in our responses to Reviewers UZen, bjw8, and adFa. A companion PDF with every revised passage rendered in blue is included as supplementary material to ease verification.

**Main changes (per reviewer):**

- **Hypothesis section renamed and reformulated** to "A Hypothesis on Mixing Redundancy": the thesis is narrowed to *explicit message-passing may duplicate the implicit mixing the router's weighted combination already performs*, with the per-token vs. across-tokens distinction made explicit and a stated falsifier (adFa concern 1; bjw8 RC2).
- **Compute honesty is now central**: §6 opens with the FLOP-matched vs. wall-clock-matched comparison table, the result is scoped precisely as a FLOP-efficiency finding, and BLEU moves to an appendix marked preliminary and is removed from the contributions (bjw8 RC3; UZen W2/W3).
- **New controls section (§6.3)**: the depth × diversity ablation (topk_depth, 10.81 ± 0.16, n = 3), dense_baseline (16.24), dense_deep (13.58), dense_16l (11.57), and depth+communication (10.67, matched-seed null) — with FFN/attention FLOP classes stated (bjw8 RC4; adFa asks 1–2).
- **Mechanism section rewritten**: the DEQ fixed-point analogy is replaced by the measured contracting, anti-aligned correction (per-layer table in Appendix C), and the "right expert" framing is softened per the routing-diffuseness results (UZen Q1–Q5, W4).
- **New supporting material**: two new appendices — per-expert input-token distributions (normalized JSD per layer; adFa ask 4) and control designs/trajectories — plus token-level attribution and per-token-id routing tables added to the existing diagnostics appendix.
- **Clarity fixes**: §3 topology-vs-payload bandwidth clarification, Algorithm 1 single-routing comments, formal entropy/load-CV definitions, self-contained introduction (adFa asks 3–5; bjw8 RC1). Claims are scoped to the tested WMT14 En-De setting throughout, and the title is revised accordingly.

**Corrections we flag transparently.** Re-verifying every number against the raw training logs while preparing the revision surfaced five items where the revised manuscript corrects the submission or our posted responses:

1. Our response tables labeled seed-42 values (A_none 15.22, S_depth 13.99) as 5-seed means; the paper's controls table uses the true means (15.08, 13.97), with seed-42 values disclosed in the caption — so "22.7%" in the response reads 22.6% in the paper.
2. topk_depth's wall-clock multiplier is ~1.13× A_none (295/262), not the ~1.16× stated in our response.
3. Table 1 row 2 (learnable topology, 15 epochs) previously reported Δ ≈ 0%, p > 0.9; the completed logs give Δ = +0.46%, p = 0.31 — verdict unchanged (not significant), and per-seed values for two screening rows were likewise corrected (their means and Δ% were already correct).
4. Two 3-seed topology controls that had been run but not included in the submitted PDF are now reported: **fixed random** and **fixed fully-connected** topologies, both directionally worse than no communication (§3, Appendix A) — completing the topology-structure axis referenced in our response to Reviewer adFa.
5. The submission described the model as "~93M total parameters"; that figure corresponds to the 8-expert configuration used in early screening. The 32-expert main model has ~290M total parameters (~268M expert weights; ~40M active per token), now stated correctly in §1, §2, §9, and Appendix B, with the ablation table's Params column clarified as expert parameters per layer. No compute-matched comparison is affected, since all FLOP matching in the paper is per-activation.

We also added two proactive clarifications: the Limitations note that training used no auxiliary load-balancing loss (a natural unrun control), and §6.2 notes that the parallel path renormalizes selected routing scores, so at top-k=1 both ablation forward paths apply the expert output at unit weight — routing is identical across the ablation conditions in the strictest sense.

The main text fits within 12 pages; references and appendices follow. We are happy to answer any further questions.

---

> ### Comment · Reviewer_adFa · 2026-07-08
> **Thanks for the work**
>
> I especially like the newly added discussion in Section 7 and the analysis in Appendix C and Appendix E.
>
> Best,